

# Development and evaluation of a Sustainable Drainage System module into TEB (v 9.0) model

José Manuel Tunqui Neira[1,2], Katia Chancibault[2], Marie-Christine Gromaire[1], Ghassan Chebbo[1]

[1]Leesu, ENPC, Institut Polytechnique de Paris, Univ Paris Est Creteil, 77455 Marne-la-Vallée, France
5  [2]GERS-LEE, Univ. Gustave Eiffel, F-44344 Bouguenais, France

*Correspondence to*: Katia Chancibault (katia.chancibault@univ-eiffel.fr)

**Abstract.** Addressing urban stormwater management challenges, this study integrates a novel Sustainable Drainage System (SUDS) module into the Town Energy Balance (TEB) model to enhance urban hydro energetic simulations. The SUDS module is developed using the *Equivalent Sustainable Drainage System* ($E - SUDS$) approach, which aggregates various SUDS based 10  on their hydrological processes and compatibility with TEB, providing a simplified representation for large-scale urban models. This study successfully develops this module, focusing on the hydrological conceptualization tailored to specific SUDS processes. A rigorous evaluation was conducted, including a comparison with the bioretention module of the SWMM model, to validate the accuracy of hydrological process dynamics and water balance simulated by the TEB SUDS module. The initial results demonstrated that the TEB SUDS module effectively simulates most of the targeted hydrological processes and 15  the key parameters involved in water balance calculations. This module offers a comprehensive tool for analysing the cumulative and spatial effects of different SUDS at an urban scale.

## 1 Introduction

Sustainable Urban Drainage System (SUDS), known by various terms across different regions (Fletcher et al., 2015), encompass a spectrum of engineered or nature-based solutions strategically placed near runoff origins to face urban stormwater 20  issues. By mimicking pre-development hydrological processes like infiltration and evapotranspiration (Mantilla et al., 2024), SUDS are integral to modern stormwater management strategies, enhancing or replacing traditional centralized networks. The strategic placement of SUDS facilities forms decentralized systems adept at directly mitigating urban runoff.

Modelling SUDS is fundamental in urban planning, offering a comprehensive approach to addressing complex stormwater management challenges, particularly those related to runoff quantity and quality issues (Wang et al., 2020). This includes 25  managing increased runoff (Mohammed and Welker, 2022; Qin et al., 2019), mitigating flood risks (Bell et al., 2020; Qin, 2020), reducing pollution of water bodies (Pennino et al., 2016; Winston et al., 2023), ensuring adequate groundwater recharge (Alamdari and Terri S, 2022; Zhang et al., 2018) and/or restoring the predevelopment hydrological balance (Chen and Chui, 2025).



Traditional hydrological models, including the *Storm Water Management Model* (SWMM;  Rossman and Huber, 2015), the

*Soil and Water Assessment Tool* (SWAT; Neitsch et al., 2011), the *Hydrologic Engineering Centre-Hydrologic Modelling System* (HEC-HMS; U.S. Army Corps of Engineers (USACE), 2000)) and the *Long-Term Hydrologic Impact Assessment Model* (L-THIA-LID, Ahiablame et al., 2012) simulate the hydrological functioning of SUDS using diverse methodologies. The SWMM model simulates SUDS using a reservoir – type approach, where a reservoir corresponds to a different vertical layer of the SUDS: surface, soil, underground storage and drainage. Surface inflow is calculated using the Manning equation,

while soil infiltration is modelled using the Green-Ampt approach (Green and Ampt, 1911). The model incorporates basic soil properties to represent water flow whitin the soil, and hydraulic characteristics are applied to simulate underdrain flow (Versini et al., 2016; Vittorio and Ahiablame, 2015). However, many models lack specific SUDS modules, requiring adaptation to represent the unique hydrological processes of SUDS. This involves defining model subcatchments as SUDS, by modifying soil and land use inputs, and adjusting key parameters such as soil hydraulic conductivity. SUDS like retention basins require

setting appropriate subcatchment retention times and outflow mechanisms (Damodaram et al., 2010; Her et al., 2017). Models like L-THIA-LID use the Curve Number approach (SCS, 1986). By modifying the CN coefficient to extend the travel time and increase the initial abstraction, the effect of SUDS on runoff generation and infiltration can be simulated (Liu et al., 2016; Wright et al., 2016). Even models with a specific SUDS model mostly lack the capability to represent the great variety of SUDS designs at the urban scale (Tunqui Neira et al., 2023) and consequently calculate their cumulative effects.

However, evaluating SUDS solely based on their stormwater management capabilities overlooks their full range of advantages. The need for a holistic assessment is heightened by the new challenges cities face, due to global warming and intense urban growth. This growth is intensifying the use of a variety of SUDS at the urban scale. Additionally, there is a growing need to understand the impact of SUDS at such a scale, necessitating models able to accurately represent their cumulative and spatial effects (Chen et al., 2023; Golden and Hoghooghi, 2018; Jefferson et al., 2017).

Global warming affects the liveability of our cities during hot periods because of urban characteristics such as building density, anthropogenic heat, material properties, and vegetation cover (Martilli et al., 2020). The use of vegetative SUDS for urban comfort is of growing interest, especially the effect of trees shading and evapotranspiration (Bevilacqua et al., 2017; Saaroni et al., 2018). However, studies on this topic remain limited. Assessing the cooling effect of vegetation and its impact on comfort at the urban scale requires simulating evapotranspiration (ET) and its interaction with surface energy balance and mass flux

transport (Robineau et al., 2022). Hydrologic SUDS models are unable to describe the various energy processes necessary to study, for example, the thermal comfort provided by SUDS. To address these new challenges, models that integrate both hydrological and energy processes are required.

Climate models may offer a solution. These models are used to simulate the surface energy balance and calculate latent heat fluxes produced by vegetation and natural surfaces. Most of these models are suitable for extensive study areas, such as a city.

However, the majority do not simulate hydrological processes (Robineau et al., 2022). One of the few models capable of simulating both hydrological and energy processes with a similar level of detail is the TEB model.



The Town Energy Balance (TEB) model (Bernard et al., 2021; Masson, 2000; Stavropulos-Laffaille et al., 2021), integrating the ISBA-DF natural transfer scheme, provides a comprehensive framework for analysing energy and water transfers in urban areas. Its mesh-based approach, focusing on urban components like buildings, roads, and green spaces, offers a nuanced view of urban hydrological processes and stands out for its integrated approach to simulating the urban environment.

This paper focuses on the development of a new SUDS module within the TEB model, employing the *Equivalent Sustainable Drainage System* ($E - SUDS$) approach. This strategic innovation aggregates diverse SUDS facilities into a unified modelling entity, simplifying their representation in large-scale urban models by grouping them based on their hydrological functions and TEB modelling environment. Our methodology begins with the implementation description of the $E - SUDS$, detailing its hydrological conceptualization and effectiveness in simulating urban hydrologic processes through two scenario analyses. The project's objective is to address existing modelling limitations by enhancing the accuracy and applicability of SUDS modelling at the urban scale and capturing their full urban management benefits, such as thermal comfort.

## 2 Strategy for the integration of SUDS module into TEB

### 2.1. Current urban hydroclimatic modelling with TEB

TEB model is integrated into the SURFEX v9.0 platform (Masson et al., 2013). Over the last decade, TEB has undergone numerous enhancements, including the integration of a garden compartment (Lemonsu et al., 2012) through coupling with ISBA (Boone et al., 2000), a natural surface scheme (SVAT – Soil-Vegetation-Atmosphere-Transfer) also part of SURFEX. Other improvements include the inclusion of trees (Redon et al., 2017), enrichment of urban water processes within ISBA under built-up surfaces, incorporation of sewer networks (Stavropulos-Laffaille et al., 2018) and the representation of green roofs (de Munck et al., 2013).

In TEB's framework, the urban environment is abstracted into an interconnected array of street canyons, considered to have infinite length. The model allows for detailed specification of the geometric, radiative, and thermal properties of an average street canyon — either representing equiprobable orientations or tailored to specific street configurations. TEB, in conjunction with the ISBA model, simulates evapotranspiration processes and hydrological dynamics, reflecting soil water status and its impact on plant transpiration capacity (Daniel et al., 2018).

TEB combines hydrological processes (drainage, infiltration, and capillary action in the soil), energetic processes (latent, sensible, and storage heat fluxes), and radiative processes (direct and diffuse solar radiation and infrared radiation) across four key urban surface types, categorized into three main compartments: "buildings" (roofs and walls), "roads," and "gardens" (encompassing vegetation and bare soil). It calculates urban microclimate parameters at the street level and assesses energy and water balances from local neighbourhoods to the broader city scale (Lemonsu et al., 2010; Schoetter et al., 2020), contributing to the understanding of urban impacts on climatic conditions when linked with an atmospheric model (Voldoire et al., 2017). In fact, TEB is one of the few models capable of performing this integrated form of modelling on a large urban scale (Robineau et al., 2022).



**2.2. Equivalent Sustainable Drainage System ($E - SUDS$) approach**

To develop the SUDS module, we have adopted the typology of SUDS for modelling purposes described by Tunqui Neira et al. (2023), which defines 16 different types of SUDSs. However, the aim in developing the SUDS module within the TEB model is to establish parameterization as simple as possible (yet effective) for the diffusion scenarios of SUDS based on the expected hydrological functions without delving into a detailed description of each SUDS. Another goal is to remain consistent with the model's level of detail and to also enhance the computation time at the mesh level. To fulfil this objective, we begin

with a comprehensive conceptualization of the hydrological processes characteristic of reservoir-type systems, tailored to the specific compartments of each SUDS among the 16 identified in the typology. This step is crucial for identifying shared processes that facilitate the aggregation of these SUDS. Subsequently, we examine the TEB modelling criteria, further refining our approach to SUDS aggregation based on these guidelines.

The integration of these methodologies enables the creation of consolidated entities within the TEB model, which amalgamate

various SUDS into cohesive units. We designate these consolidated entities as *Equivalent Sustainable Drainage System (E − SUDS)*. This strategic assembly not only streamlines the modelling process but also enhances the representation and analysis of SUDS in urban hydrological studies.

**2.2.1 Conceptualisation of the hydrological processes of the SUDS typology**

*Note:* from now on, to refer to one of the 16 SUDSs in the typology (Figure 1), we will use the following nomenclature: the

letter $R$ followed by the row number ($R\#$) and the letter $C$ with the column number ($C\#$) where the SUDS is located. Thus, for example, the SUDS $R1C2$ refers to the SUDS formed by the structure type: open air – percolation through a substrate ($O_{PE}$, Figure 1) and the main SUDS' hydrological assured function: runoff volume reduction – evapotranspiration ($V_{EV}$, Figure 1).

The conceptualization of the hydrological processes within the 16 groups of SUDS of the typology makes it possible to define six types of hydrological reservoirs which are (Figure 2 and Figure 3):

• **Storage – Infiltration reservoir (SI):** It conceptualizes the temporary retention of runoff on the surface of the SUDS followed by its infiltration into the soil of the SUDS. This reservoir represents the *temporary surface water compartment* in the typology (Figure 1). It collects runoff from surfaces connected to the SUDS plus direct precipitation falling on the SUDS's surface. The stored water is mainly evacuated through infiltration into the SUDS's soil, and to a lesser extent by evaporation. Excess water when the reservoir is full undergoes overflow.

• **Constant water reservoir (CW)**: This reservoir is permanently connected to the storage-regulation reservoir (described below) and represents the *permanent surface water compartment* in the typology (Figure 1). This reservoir is supplied in water by the storage-regulation reservoir. The CW reservoir is fed by the storage-regulation reservoir, and the water stored in it can only be evacuated through evaporation.

• **Transport – Infiltration reservoir (TI):** It conceptualizes the temporary retention of rain/runoff on the surface of

the SUDS during its transport by surface runoff to an outlet or another SUDS, and (if conditions allow) also its



infiltration into the soil of the SUDS. This reservoir represents the temporary water compartment for the transport in the SUDS typology (Figure 2). This reservoir collects the water flux from precipitation falling directly onto the SUDS's surface and the runoff from surfaces connected to the SUDS. The stored water is then channelled along the SUDS to an outlet/another SUDS, infiltrated into the SUDS's soil, evaporated. A portion of this water may also be evacuated by the potential overflow of the reservoir.


- **Storage-Exfiltration reservoir (SE):** It conceptualizes the temporary retention of rainwater in the underground storage compartment of the SUDS, followed by its exfiltration to the natural soil surrounding the SUDS. This reservoir can represent 2 cases (Figure 1): 1) The *storage/exfiltration compartment* of the typology; 2) The part below the underdrain of the *storage/drainage/exfiltration compartment* of the SUDS typology. This reservoir primarily collects water flux from the *substrate compartment* of the SUDSs in the typology. The stored water is then evacuated through exfiltration to the natural soil beneath the SUDS or by potential overflow of the reservoir (only case 1). To represent case 1, the SE reservoir operates alone. For case 2, the SE reservoir is connected to the storage-regulation reservoir.


- **Storage – Regulation reservoir (SR):** It conceptualizes the temporary retention of rainwater followed by its discharge, via a flow-regulated device, into the sewer network. This reservoir can represent 3 cases (Figure 1): 1) The *temporary surface water compartment with flow regulation device* in the typology; 2) The part above the underdrain of the *storage/drainage/exfiltration compartment* in the typology; 3) The *lined storage/drainage compartment* in the typology. This reservoir can collect surface precipitation/runoff (case 1), and percolation from the *substrate compartment* of the SUDS (cases 2 and 3). In case 1, the SR reservoir can supply water into a SI reservoir or the CW reservoir. In case 2, the SR reservoir supplies water into a SE reservoir. Flow regulated outflow only happens once these underlying SI, CW or SE reservoirs are full. Finally, for case 3, the SR reservoir operates independently. Excess water can also be evacuated by overflow of the reservoir.



- **Soil reservoir:** The soil reservoir is designed to simulate the dynamics of water flow through the soil layer within a SUDS framework. To enhance practical application and accuracy, this module adopts the conceptual foundation provided by the ISBA-DF model (Boone et al., 2000; Decharme et al., 2011), which proficiently simulates water flux within the soil across the tripartite urban compartments delineated by the TEB model (i.e., building, road and garden). The hydrological process within the ISBA-DF framework is articulated through an implementation of the Richards equation in its "mixed" form (Brooks and Corey, 1966), facilitating an advanced depiction of soil water mass transfer pursuant to Darcy's law. This methodological approach enables the resolution of water movement dynamics by calculating volumetric water content and applying hydraulic gradients expressed as water pressure heads.







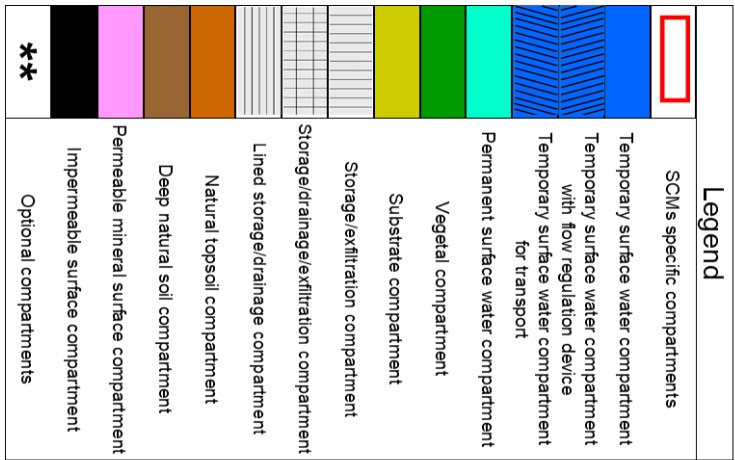

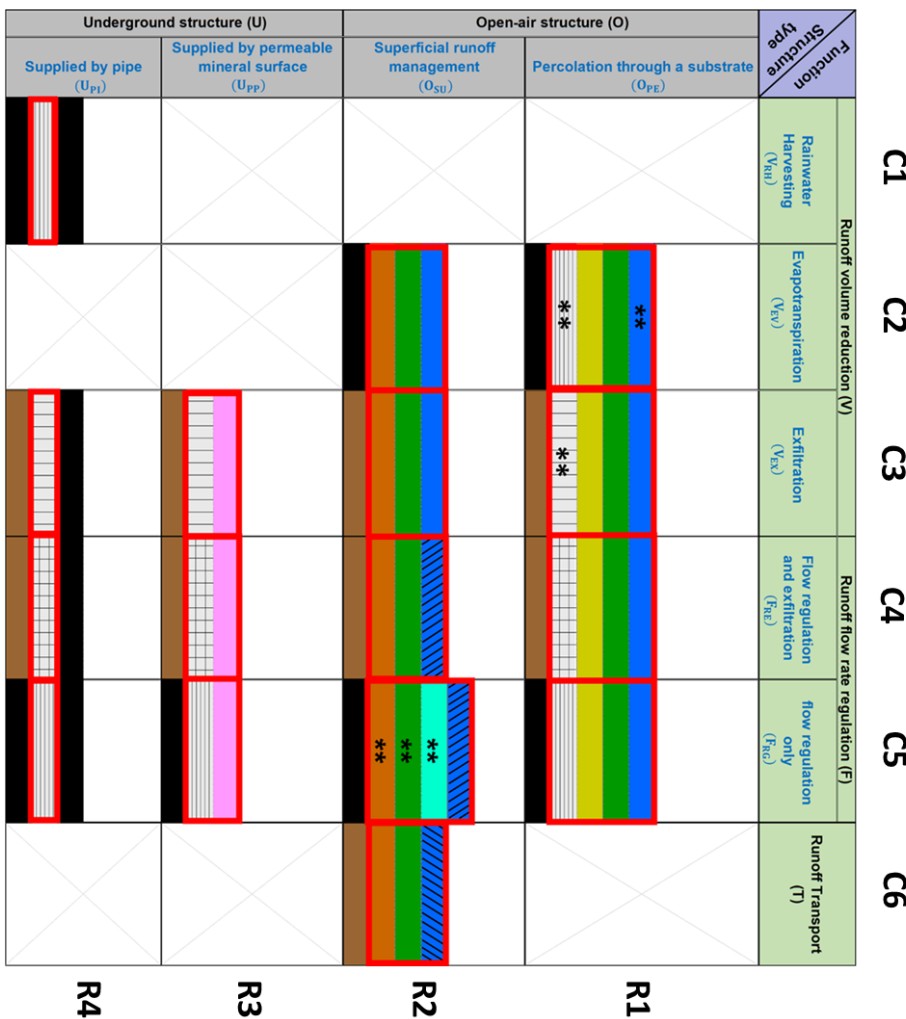

**Figure 1: SUDS typology proposed by Tunqui Neira et al. (2023)**





To support a comprehensive analysis of soil thermal and moisture dynamics, soil temperature and moisture profiles are cohesively calculated across a vertically distributed grid comprising distinct soil layers, each with variable thicknesses that extend to the maximum depth of the selected substrate. This vertical stratification is crucial for the precise calculation of soil thermal parameters, which are intrinsically dependent on the hydrological properties of each soil layer. Moreover, to ensure the accurate representation of moisture content beyond the root zone, soil moisture for each subsequent deeper layer is calculated based on the equilibrium between gravitational and capillary forces as postulated in Darcy's law. This calculation is pivotal for maintaining a balanced representation of soil water content, thereby facilitating a more refined and realistic simulation of subsurface hydrological processes within urban ecosystems. This reservoir represents the *substrate* compartment of the SUDSs in the typology (Figure 1).

Figure 2 illustrates the disposition of the reservoirs in SUDS $R1C3$, $R1C4$ and $R1C5$ of the typology.

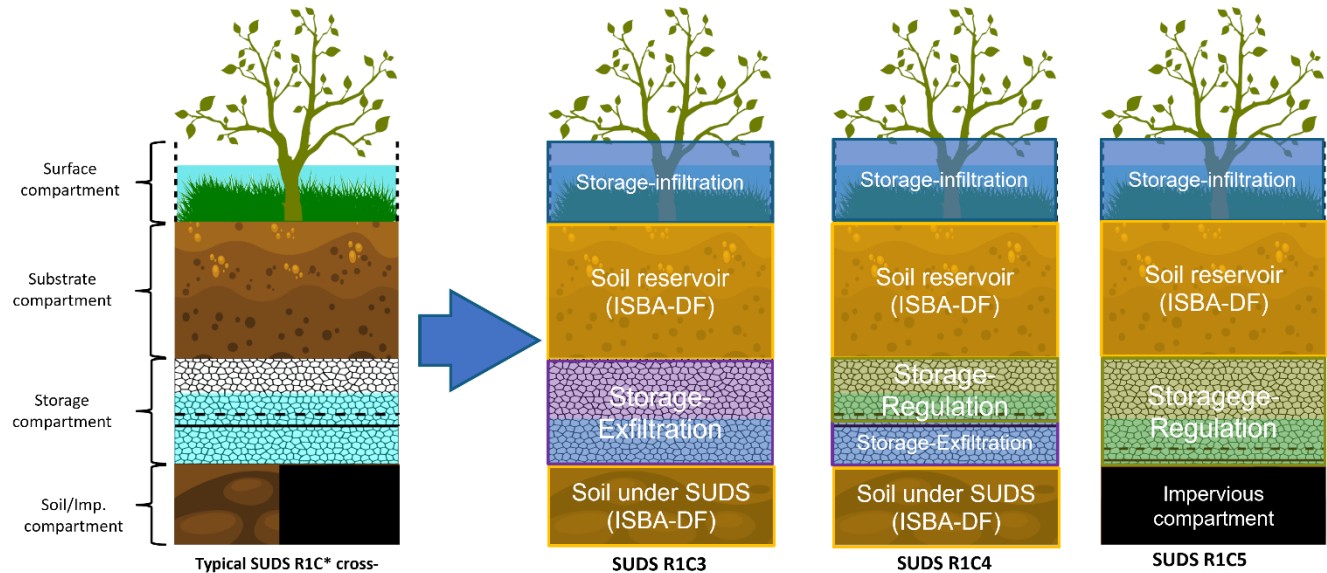

**Figure 2: Example of the disposition of reservoirs in SUDS R1C3, R1C4 and R1C5 of the typology**

**2.2.2 TEB modelling criteria**

Beyond hydrological conceptualization, the development of the $E - SUDS$ method necessitates to consider specific TEB modelling constraints. Here are five pivotal constraints to guide this process:

1. **Urban landscape composition:** This includes the integration within the model's urban compartments, such as buildings, roads, and gardens.

2. **Vegetation-Soil hydro-energetic processes:** This refers primarily to whether the SUDS includes vegetation, as this significantly influences the interaction between soil moisture, energy exchange, and hydrological behaviour.

3. **SUDS and Urban canyon surface interaction:** How the SUDS interacts with the TEB model's canyon street surface, impacting the model's comprehensive hydro-energetic balance.



4. **Soil layer discretization continuity:** Critical for a model as ISBA-DF that simulates soil layer processes, ensuring seamless integration of the SUDS within the soil's discretization framework.

5. **Water transport dynamics:** The mechanism by which the SUDS manages the collection and subsequent transport of water.

In our preliminary approach to modelling within TEB, in order to streamline programming and model configuration efforts, we will proceed under the assumption that the hydraulic properties of the SUDS's substrate do not significantly deviate from those of the model's surface soil layer. This simplification allows us to avoid distinguishing between SUDSs categorized by "substrate" versus those identified by "soil» and enables us to maintain lateral water fluxes between the substrate layers of the SUDS and the soil layers of other compartments in the mesh (Stavropulos-Laffaille et al, 2018).

The combination of the hydrological conceptualisation plus the TEB modelling configuration criteria results in the identification of five $E - SUDS$ types which are summarised in Table 1

Table 1: Summary table presenting the type of $E - SUDS$, the SUDSs and reservoirs that comprise it, based on Figure 1Figure 2

| $E - SUDS$ type | SUDS that form the $E - SUDS$ | Reservoirs used by the $E - SUDS$ |
|---|---|---|
| $E - SUDSa$ | $R1C2$ (with and without underground storage) $R1C3, R1C4$ and $R1C5$ $R2C2$ $R2C5$ (vegetated without superficial permanent water) ° | Storage – Infiltration (SI) Storage – Exfiltration (SE) Storage – Regulation (SR) Soil |
| $E - SUDSb$ | $R2C3$ (without underground storage) $R2C4$ | Storage – Infiltration (SI) Storage – Exfiltration (SE) Soil |
| $E - SUDSc$ | $R2C6$ | Transport – Infiltration (TI) |
| $E - SUDSd$ | $R3C3, R3C4, R3C5$ $R4C3, R4C4$ | Storage – Exfiltration (SE) Storage – Regulation (SR) Soil |
| $E - SUDSe$ | $R2C5$ (non-vegetated with superficial permanent water) $R2C5$ (non-vegetated without superficial permanent water) | Constant water (CW) Storage – Regulation (SR) |

The $E - SUDSa$, $E - SUDSb$ and $E - SUDSc$ types cover the vegetated SUDSs within the typology, exhibiting hydro-energetic dynamics comparable to the garden compartment of the TEB mesh. The $E - SUDSd$ type includes all underground



SUDS in the typology, mirroring the hydro-energetic behaviour of the TEB model's road compartment. Lastly, the $E - SUDSe$
type consists of surface-level, lined, non-vegetated SUDS. Similar to $E - SUDSd$, $E - SUDSe$ shares hydro-energetic
characteristics with the TEB model's road compartment.

In the modelling process, these five $E - SUDS$ types each occupy a specific portion of the mesh Figure A.1 in Appendix A),
enabling the distinction of their unique hydro-energetic outputs in comparison to the three original compartments of the mesh
(building, road, garden). This spatial allocation facilitates a nuanced analysis of the varied hydro-energetic contributions made
by each $E - SUDS$ type within the urban landscape. In this revised mesh configuration, the five $E - SUDS$ types will collect
runoff from the impervious surfaces of the TEB mesh (i.e., building and roads) that do not directly link to the TEB sewer
network.

Finally it is important to note that none of the five $E - SUDS$ types incorporate the green roof SUDS type, as there is already
an existing module within the TEB framework specifically designed to simulate this type of SUDS (de Munck et al., 2013).

**3 Hydrological functioning of the Equivalent Sustainable Drainage System approach (Figure 3)**

In this article, we provide a detailed explanation of the hydrological functioning of the $E - SUDSa$ type, which stands out as
the most complex among the five types, incorporating nearly all aspects of reservoir dynamics. The functioning of the
remaining four types will be detailed in the forthcoming SURFEX V9 scientific documentation, published by the CNRM-
GAME research team. Their evaluation will be further published. However, the conceptualisation of the hydrological
functioning of these four $E - SUDS$ is presented in Appendix B of this paper.

In the $E - SUDSa$ type there can be SUDSs where the surface compartment is represented by the combination of two
reservoirs: storage-infiltration (SI) and storage-regulation (SR) (SUDS $R2C5$, Table 1), or only by the storage-infiltration
reservoir (SUDS $R1C2$, $R1C3$, $R1C4$, $R1C5$, and $R2C2$, Figure 2 and Table 1). When SUDS $R2C5$ is combined with other
SUDS, runoff from impervious surfaces connected to $R2C5$ must first enter to SR reservoir before being discharged into the
SI reservoir (represented by $F_{SR-SI}(t)$). This configuration is designed to eliminate backflow between the SI and SR reservoirs,
optimizing computation time. Additionally, this $E - SUDS$ type may feature SUDS with an underground storage system
depicted by either a combination of storage-exfiltration (SE) and storage-regulation (SR) reservoirs ($R1C4$, Figure 2 and Table
1) or exclusively by a storage-regulation reservoir ($R1C5$, Figure 2 and Table 1). In combined reservoir setups, all water flux
from the soil reservoir will first enter the SR reservoir and then discharge into the SE reservoir (represented by $F_{SR-SE}(t)$),
preventing backflow between SE and SR.

**3.1. Functioning of the Storage – Infiltration reservoir (SI)**

The SI reservoir operates based on the water balance equation, which is formulated as follows:

$$\frac{\partial W_{SI}}{\partial t} = f_{SI} \cdot [P_{SI}(t) + R_{SI}(t)] - F_{SI-ovf}(t) + F_{SR-SI}(t) - F^*_{SI-evp}(t) - F^*_{SI-inf}(t), \quad\quad (1)$$





where $W_{SI}$ is the water level stored in the SI reservoir (kg·m$^{-2}$), $P_{SI}(t)$ the direct rainfall (kg·m$^{-2}$·s$^{-1}$), $R_{SI}(t)$ the collected runoff (kg·m$^{-2}$·s$^{-1}$), $f_{SI}$ is the fraction of SUDSs of the $E-SUDSa$ type equipped with only the SI reservoir at the surface (-),

$F_{SI-ovf}(t)$ the overflow from the SI reservoir (kg·m$^{-2}$·s$^{-1}$), $F_{SR-SI}(t)$ the water flux from the storage – regulation reservoir (kg·m$^{-2}$·s$^{-1}$), $F^*_{SI-evp}(t)$ the evaporation flux (kg·m$^{-2}$·s$^{-1}$) and $F^*_{SI-inf}(t)$ the infiltration flux directed towards the substrate of the $E-SUDSa$ type (kg·m$^{-2}$·s$^{-1}$).

The $f_{SI}$ fraction can be calculated as follows:

$$f_{SI} = \frac{\sum_{R1C2,R1C3,R1C4,R1C5,R2C2} S_{imp}}{\sum_{R1C2,R1C3,R1C4,R1C5,R2C2,R2C5} S_{imp.}}, \tag{2}$$

where $S_{imp}$ is the impervious surfaces linked to concerned SUDS (m$^2$). A preparatory assessment is crucial to identify the

impervious zones within the TEB mesh that connect with each participating SUDS in this segment before starting the $E-SUDS$ modelling.

The overflow $F_{SI-ovf}(t)$ can be calculated as follows:

$$F_{SI-ovf}(t) = \frac{\max[0; W_{SI}(t) - W_{SIMAX}]}{\Delta t}, \tag{3}$$

where $W_{SIMAX}$ is the maximum level of the storage-infiltration reservoir (kg·m$^{-2}$) and $\Delta t$ is the time step (s). The overflow is sent as runoff to the TEB garden compartment.

The water flux from the SR reservoir can be calculated as follows:

$$F_{SR-SI}(t) = \frac{min[(W_{SIMAX} - W_{SI}(t)); W_{SR}(t-1) + (1 - f_{SI}) \cdot [P_{SR}(t) + R_{SR}(t)] \cdot \Delta t]}{\Delta t}, \tag{4}$$

where $W_{SR}(t-1)$ is the water level in the storage-regulation reservoir at time step $t-1$ (kg·m$^{-2}$), while $P_{SR}(t)$ and $R_{SR}(t)$ the direct rainfall and runoff (kg·m$^{-2}$·s$^{-1}$) respectively, at the current time step.

In the TEB model, water evaporation from vegetated surfaces is only considered when surface overflow occurs due to the saturation of the soil column in the garden compartment. (Decharme et al., 2011, 2012). Thus, this process has been adapted

for the reservoirs (SI and SR) as follows:

$$F_{evp}(t) = \frac{\left(\frac{W_*(t)}{W_{*MAX}}\right)^{2/3} \cdot \rho_a \cdot C_H \cdot V_a \cdot [q_{sat}(T_s) - q_a]}{\Delta t}, \tag{5}$$

where (*) stands for the SI or SR reservoirs, $F_{evp}(t)$ represents the water evaporation flux (kg·m$^{-2}$·s$^{-1}$), $\rho_a$ the air density (kg·m$^{-3}$), $C_H$ the dimensionless drag coefficient depending upon the thermal stability of the atmosphere, $q_a$ the air humidity (kg kg$^{-1}$), $V_a$ the wind speed (m·s$^{-1}$) and $q_{sat}$ the saturated specific humidity at the surface (kg kg$^{-1}$) that depends on surface temperature, $T_s$ (°K).

In the SI reservoir, the $F^*_{SI-evp}(t)$ flux can be defined as follows:

$$F^*_{SI-evp}(t) = min\left[\frac{W_{SI}(t)}{\Delta t}, F_{evp}(t)\right]. \tag{6}$$





**Figure 3: Conceptualisation of the hydrological functioning of the $E - SUDSa$ type**





The evaporated water flux will then be aggregated with the evapotranspiration flux from the canyon street of the TEB model
for the corresponding energy/water balance. Appendix C provides a more detailed explanation of the development of this
equation.

The infiltration flux $F^*_{SI-inf}(t)$ is calculated using the default function from the ISBA-DF model (Albergel et al., 2018), based
on the potential (supply limited) infiltration rate ($I_c$ in kg·m⁻²·s⁻¹). This function follows the soil infiltration approach proposed
by Hewlett and Hibbert (1967), which assumes that the soil saturation progresses from the bottom of the column upward. As
a result, water continues to infiltrate until the entire soil column is fully saturated, after which runoff occurs at the soil surface.
Therefore, within the reservoir, this flux can be defined as follows:

$$F^*_{SI-inf}(t) = min\left[\frac{W_{SI}(t)}{\Delta t}, I_c(t)\right],\qquad(7)$$

$$I_c(t) = \rho_\omega \cdot \sum_{j=1}^{nb_{subs}} \left[\frac{(\omega_{sat,j} - \omega_{l,j}) \cdot \Delta z_j}{\Delta t}\right],\qquad(8)$$

where $\rho_\omega$ is the water density (1000 kg·m⁻³), $nb_{subs}$ the total number of soil layers in the SUDS substrate, $\omega_{sat,j}$ the soil
porosity (m³·m⁻³) of the $j$-th soil layer in the substrate, $\omega_{l,j}$ the current soil water content (m³·m⁻³) and $\Delta z_j$ the thickness of the
soil layer (m).

The infiltration flux is transferred to the $E - SUDSa$ soil reservoir, which is managed by the ISBA-DF model.

**3.2. Soil reservoir (substrate, below the SI reservoir)**

Here, everything related to water flux in the soil (i.e., different discretized soil layers) is simulated after the ISBA-DF model.
The water flux in the last discretized soil layer (in kg·m⁻²·s⁻¹), denoted as $F^*_{perc}(t)$ (gravitational drainage; $F_N(t)$ in Albergel
et al. (2018)), represents the percolation flux between the soil and the underground storage layer of the $E - SUDSa$. Although
there is a soil capillary break due to the placement of an underground storage compartment in the $E - SUDSa$, theoretically,
a seepage flow boundary condition should be imposed to activate $F^*_{perc}(t)$ (for example, based on the soil's matric potential).
However, it has been decided to adopt a simple formulation for this process which is evaluated in this paper. Thus, it can be
stated that $F^*_{perc}(t) = F_N$. In the ISBA-DF model, the default seepage flow boundary condition for $F_N(t)$ is of the *free
drainage* type (hydraulic gradient equal to 1). Thus $F_N(t) = -K_N(t)$ (i.e., the hydraulic conductivity in the bottom soil layer
of the SUDS substrate).

When $F^*_{perc}(t) \neq 0$, this water flux must then be distributed among the two reservoirs that represent the underground storage
reservoir of $E - SUDSa$ (i.e., storage-exfiltration, storage-regulation), and a non-limiting drain outlet, but first, the amount
corresponding to SUDS $R2C2$ and $R2C5$, which are completely lined structures, must be subtracted from $F^*_{perc}(t)$. For this,
we apply the following relationship:

$$F^*_{perc-l}(t) = f_{ln} \cdot F^*_{subs}(t),\qquad(9)$$



where $F^*_{perc-l}(t)$ is the potential percolation flux generated by the lined SUDS (kg·m$^{-2}$·s$^{-1}$), and $f_{ln}$ the proportion of lined

SUDSs classified under the $E - SUDSa$ type. This value is determined using the following calculation:

$$f_{ln} = \cdot \frac{\sum_{R2C2,R2C5} S_{imp}}{\sum_{R1C2,R1C3,R1C4,R1C5,R2C2,R2C5} S_{imp.}}. \qquad (10)$$

The updated percolation flux is calculated as: $F^*_{perc}(t) = F^*_{perc}(t) - F^*_{perc-l}(t)$. Subsequently, $F^*_{perc-l}(t)$ is converted

into soil water content (m$^3$·m$^{-3}$) and then reintroduced into the bottom layer of the soil reservoir.

Finally, the distribution of the percolation flux among the reservoirs is as follows:

$$F_{perc-SE}(t) = f_{SE} \cdot F^*_{perc}(t), \qquad (11)$$
$$F_{perc-SR}(t) = f_{SR-SE} \cdot F^*_{perc}(t), \qquad (12)$$
$$F^*_{dr-n}(t) = (1 - f_{SE} - f_{SR-SE}) \cdot F^*_{perc}(t), \qquad (13)$$

where $F_{perc-SE}(t)$ is the percolation flux directed to the storage-exfiltration reservoir (kg·m$^{-2}$·s$^{-1}$), $F_{perc-SR}(t)$ the percolation

flux directed to the storage-regulation reservoir (kg·m$^{-2}$·s$^{-1}$), and $F^*_{dr-n}(t)$ the percolation flux evacuated by the non-limiting

drain into the TEB sewer network (kg·m$^{-2}$·s$^{-1}$). The term $f_{SE}$ specifies the proportion of SUDS within the $E - SUDSa$ that

possess only an underground SE reservoir (-), while $f_{SR-SE}$ specifies the proportion of SUDS within the $E - SUDSa$ that

incorporate both SE and SR reservoirs to depict their underground storage compartment (-). The fractions $f_{SE}$ and $f_{SR-SE}$ are

calculated as follows:

$$f_{SE} = \cdot \frac{\sum_{R1C3} S_{imp}}{\sum_{R1C2,R1C3,R1C4,R1C5} S_{imp.}}. \qquad (14)$$

$$f_{SR-SE} = \cdot \frac{\sum_{R1C4,R1C5} S_{imp}}{\sum_{R1C2,R1C3,R1C4,R1C5} S_{imp.}}. \qquad (15)$$

The evapotranspiration in the soil reservoir $F^*_{sol-tp}(t)$ (kg·m$^{-2}$·s$^{-1}$) is calculated considering the various contributions from

vegetation and soil (Stavropulos-Laffaille et al., 2018):

$$F^*_{sol-tp}(t) = F^*_{veg-etp}(t) + F^*_{gr-evp}(t) + F^*_{gri-evp}(t) + F^*_{snow-sbn}(t), \qquad (16)$$

where $F^*_{veg-etp}(t)$ is the vegetation evapotranspiration, $F^*_{gr-evp}(t)$ and $F^*_{gri-evp}(t)$ the evaporation from the bare soil,

respectively, with and without freezing and $F^*_{snow-sbn}(t)$ the sublimation from the snow. These terms are detailed in the

SURFEX scientific documentation (Albergel et al., 2018)

**3.3. Functioning of the Storage – Exfiltration reservoir (SE)**

The SE reservoir operates based on the water balance equation, which is formulated as follows:

$$\frac{\partial W_{SE}}{\partial t} = F_{perc-SE}(t) - F_{SE-ovf}(t) + F_{SR-SE}(t) - F^*_{SE-ex}(t), \qquad (17)$$

where $W_{SE}$ represents the water level stored in the SE reservoir (mm), $F_{perc-SE}(t)$ the percolation flux from the substrate

(kg·m$^{-2}$·s$^{-1}$), $F_{SE-ovf}(t)$ the overflow from the SE reservoir (kg·m$^{-2}$·s$^{-1}$), $F_{SR-SE}(t)$ the water flux from the storage – regulation

reservoir (kg·m$^{-2}$·s$^{-1}$), and $F^*_{SE-ex}(t)$ the exfiltration flux to the natural soil beneath the $E - SUDSa$ type (kg·m$^{-2}$·s$^{-1}$). The



design presupposes that vegetation roots within the substrate do not penetrate this compartment, thus excluding the occurrence of any evapotranspiration processes.

The process for computing $F_{perc-SR}(t)$ is outlined in Eq ( 11 ).

The overflow $F_{SE-ovf}(t)$ can be calculated as follows:

$$F_{SE-ovf}(t) = \frac{max[0; W_{SE}(t) - W_{SEMAX} \cdot \emptyset_{SE}]}{\Delta t}, \tag{18}$$

where $W_{SEMAX}$ is the maximum level of the storage-exfiltration reservoir (mm) and $\emptyset_{SE}$ is the porosity or void ratio of the underground material. The overflow is sent as runoff to the TEB garden compartment.

The water flux from the SR reservoir can be calculated as follows:

$$F_{SR-SE}(t) = \frac{min[(W_{SEMAX} \cdot \emptyset_{SE} - W_{SE}(t)); W_{SR}(t) + F_{perc-SE}(t) \cdot \Delta t]}{\Delta t}, \tag{19}$$

where $W_{SR}(t)$ represents the water level in the storage-regulation reservoir at the current time step (mm).

The exfiltration flux $F^*_{SE-ex}(t)$ towards the soil beneath the $E - SUDSa$, is calculated based on the exfiltration rate ($F_{exf}(t)$

in kg·m⁻²·s⁻¹) described in Błażejewski et al. (2018), which also considers, in addition to the bottom, the exfiltration from the lateral walls of the underground storage compartment—very important in exfiltration-type SUDS. $F_{exf}(t)$ can be expressed as follows:

$$F_{exf}(t) = \rho_\omega \cdot k_{sat} \cdot \frac{\left[2,1466 + \left(\frac{b_{SE}}{W_{SE}(t)}\right)^{0,77}\right]^{1,3}}{2 + \frac{b_{SE}}{W_{SE}(t)}}, \tag{20}$$

where $k_{sat}$ is the soil's hydraulic conductivity at saturation (m·s⁻¹), and $b_{SE}$ is the width of the underground storage compartment (mm).

Therefore, in the SE reservoir, $F^*_{SE-ex}(t)$ can be defined as follows:

$$F^*_{SE-ex}(t) = min\left[\frac{W_{SE}(t)}{\Delta t}, F_{exf}(t)\right]. \tag{21}$$

The exfiltration flux is directed to the natural soil column under the $E - SUDSa$ managed by the ISBA-DF model .

### 3.4. Functioning of the Storage – Regulation reservoir (SR)

The SR reservoir operates based on the water balance equation, which is formulated as follows:

$$\frac{\partial W_{SR}}{\partial t} = (1 - f_{SI}) \cdot [P_{SR}(t) + R_{SR}(t)] - F_{SR-SI}(t) - F^*_{SR-evp}(t) + F_{perc-SR}(t) - F_{SR-SE}(t) \\ - F_{SR-ovf}(t) - F^*_{SR-dr}(t), \tag{22}$$

where $W_{SR}$ represents the water level stored in the SR reservoir (kg·m⁻²), $F^*_{SR-evp}(t)$ the evaporation flux (kg·m⁻²·s⁻¹), $F_{SR-ovf}(t)$ is the overflow from the SR reservoir (kg·m⁻²·s⁻¹), $F^*_{SR-dr}(t)$ denotes the regulated outflow (kg·m⁻²·s⁻¹). The rest of the variables in Eq. ( 22 ) have been previously defined.





The formula for $F_{SR-SI}(t)$, as detailed in Eq. ( 4 ), calculates the water flux towards the storage-infiltration reservoir

In the SR reservoir, the $F^*_{SR-evp}(t)$ flux, as in the case of SI reservoir can be calculated from the water evaporation formula

(Eq. ( 5 )):

$$F^*_{SR-evp}(t) = min\left[\frac{W_{SR}(t)}{\Delta t}, F_{evp}(t)\right]. \tag{23}$$

The evaporated water flux will then be aggregated with the evapotranspiration flux from the canyon street of the TEB model

for the corresponding energy/water balance.

The process for computing $F_{perc-SR}(t)$ is outlined in Eq. ( 12 ). The process for computing $F_{SR-SE}(t)$ is outlined in Eq. ( 19

). The overflow $F_{SR-ovf}(t)$ can be calculated as follows:

$$F_{SR-ovf}(t) = \frac{max[0; W_{SR}(t) - W_{SRMAX}]}{\Delta t}, \tag{24}$$

where $W_{SRMAX}$ is the maximum level of the SR reservoir (kg·m$^{-2}$). The overflow is sent as runoff to the TEB garden

compartment.

The regulated outflow is derived from the equation proposed by Sage et al. (2024), formulated as follows:

$$F^*_{SR-dr}(t) = F_{max(SR-dr)} \cdot \left[a_1 + (1 - a_1) \cdot \left(\frac{W_{SR}(t)}{W_{SRMAX}}\right)^{a_2}\right], \tag{25}$$

where $F_{max(SR-dr)}$ is the maximum regulated outflow linked to the TEB sewer network (kg·m$^{-2}$·s$^{-1}$); $a_1$ is the coefficient

determining the water flux for low flows (-), $a_2$ is the coefficient influencing the rate of reaching $F_{max(SR-dr)}$(-).

**3.5. Natural soil column under $E - SUDSa$**

The exfiltration flux produced in the SE reservoir is connected to simulate water flow in the soil surrounding the SUDS. The

equations governing water flux in the soil are identical to those of the ISBA-DF model. Here, we assume that the initial upper

boundary condition of the soil column for exfiltration is the same as the substrate surface condition for infiltration. This

assumption accounts for the presence of trapped air in the underground storage compartment, which mirrors the conditions at

the SUDS surface. Additionally, it is assumed that plant roots do not reach the underground storage and that its base is not

directly exposed to the atmosphere. Consequently, evapotranspiration in the soil column is considered negligible. These

assumptions are adopted by different models working with exfiltrating SUDS (e.g., Braga et al., 2007; Lee et al., 2015).

**3.6. Lateral soil water transfer between the $E - SUDSa$ and the others TEB mesh compartments**

Lateral water transfer (Bernard et al., 2021; Stavropulos-Laffaille et al., 2018) interactions from each soil layer, originating

from the three urban compartments of the TEB mesh (building, road, and garden) plus the $E - SUDSa$, are taken into account.

Structural layers, such as roadways and underground storage compartments of the $E - SUDSa$, are not included in the

calculation of horizontal transfer. It is important to note that this calculation is performed at the end of the numerical time step,

that is, after calculating the vertical flux individually in each urban compartment of the TEB mesh.





This approach is based on the principle of exponential decay of water content, trending towards the average soil moisture of
the four compartments, which is limited by the soil water content at the wilting point. Updating the soil water content in each
layer and compartment after each time step allows for:

$$\omega_{*lj}{'} = \omega_{*lj}{'} + (\overline{\omega}_{*lj} - \omega_{*lj}) \cdot \left(1 - \exp\left[-\left(\frac{\Delta t}{30 \cdot \tau}\right)\right]\right), \tag{26}$$

with

$$\overline{\omega}_{*lj} = \frac{\sum \omega_{*lj} \cdot f_*}{\sum f_*}. \tag{27}$$

The asterisk (*) refers to the application of the different terms to the four different compartments, respectively garden,
buildings, roads and $E - SUDSa$. The terms $\omega_{*lj}$ and $\omega_{*lj}{'}$ are the soil water content for each compartment, respectively,
before and after horizontal balancing (m³·m⁻³), $\overline{\omega}_{*lj}$ is the mean soil water content of all compartments before balancing (m³·m⁻³), $\tau$ the time constant for 1 day, $\Delta t$ the numerical time step and $f_*$ the fraction of each compartment in the TEB mesh.

## 4  $E - SUDSa$ hydrological evaluation

The methodology for evaluating the $E - SUDSa$'s hydrological conceptualization is based on two scenarios:

- Scenario 1: Comparison of SWMM/LID and TEB $E - SUDSa$ for a single facility to evaluate the dynamics of
hydrological processes and resulting water balance.
- Scenario 2: Combining multiple SUDS facilities, of a same type, within the $E - SUDSa$, to evaluate the equivalent
    concept developed in this study. For this scenario, only the TEB model has been used.

The two scenarios are detailed below.

### 4.1. Scenario 1: Comparison of SWMM/LID and TEB / $E - SUDSa$ for a single facility

For this scenario, the $E - SUDSa$ modelled in TEB is based on a single SUDS facility, belonging to type $R1C3$ (Figure 1).
This design includes a surface Storage-Infiltration (SI) reservoir coupled with a soil reservoir that simulates the substrate layer,
and a Storage-Exfiltration (SE) reservoir positioned above the soil column beneath the SUDS structure (as illustrated in Figure
2). The hydrological behaviour of this configuration is compared with that of the LID bioretention module of the SWMM
model (Rossman and Huber, 2015). This comparison is carried out by ensuring an identical configuration for both systems to
guarantee a fair and accurate evaluation. However, the aim of the comparison is to assess the adequacy of the approach used
to conceptualize $E - SUDSa$ within the TEB model, rather than to determine which of the two models provides more accurate
simulations.




### 4.1.1 Parameterisation of Scenario 1 in TEB model

For this study, the TEB mesh represents an urban catchment created from data provided by an existing catchment: Pin Sec in

Nantes (Stavropulos-Laffaille et al., 2021), covering 1 hectare with land use allocations of gardens (22%), buildings (50%), and roads (28%). Impervious surfaces depression water storage has a maximum of 2mm and no infiltration through pavement. For this study, the runoff produced by the impervious surfaces is entirely redirected to $E - SUDSa$.

In the TEB mesh, the soil is uniformly composed of 51% sand, 41% silt, and 8% clay across 12 layers to a depth of 3 meters. This specific texture composition allows for the application of pedotransfer functions to accurately determine soil water

dynamics within the model (Cosby et al., 1984). Gardens predominantly feature low vegetation (95%) with an adapted monthly LAI. Meteorological data, sourced from Pin Sec and Nantes Airport, feed the model, with hourly updates and a numerical resolution of 5 minutes from May 2010 to August 2012 (Stavropulos-Laffaille et al., 2021).

For the sizing of the $E - SUDSa$ surface area, the Oasis SUDS tool (Sage et al., 2024) was used. This tool requires input parameters such as the annual runoff interception target of the SUDS, the depth of the SUDS surface reservoir and the hydraulic

conductivity of the SUDS substrate to determine the appropriate surface area. For the $E - SUDSa$, the design was based on intercepting 80% of annual runoff – this proportion corresponds to the typical runoff reduction achieved during moderate rainfall events (Tunqui Neira et al., 2023) – with a surface reservoir height of 100 mm and a substrate hydraulic conductivity of 24.9 mm·h⁻¹. According to the Oasis SUDS tool, for an infiltration facility in open ground, this correspond a surface area of 238.4 m² (or a fraction of 0.024). Although the OASIS tool was originally developed for sizing SUDS areas with a

continuous soil column (i.e., without an underground storage compartment), it can still provide an estimate of the surface area required for the $E - SUDSa$. The substrate thickness is set at 0.6 m (i.e., 8 discretized soil layers), determined through literature references (Flanagan et al., 2017; Huang et al., 2025; Li et al., 2021), and the vegetation configuration mirrors that of the garden areas. The maximum height of the surface reservoir (SI) is set to 100 mm. The underground storage's maximum height (SE) is 400 mm, with a porosity of 0.4 and an exfiltration width of 10.0 m for lateral water flux. As with the substrate, these

dimensions were defined based on literature sources (Flanagan et al., 2017; Huang et al., 2025; Li et al., 2021).

Finally, to achieve the most accurate comparison possible with the SWMM model, we have chosen to disable the horizontal soil water transfers between the $E - SUDSa$ compartment and the other compartments of the TEB mesh (roads, buildings, and gardens).

### 4.1.2 Parameterisation of Scenario 1 in SWMM model

Parameters for the SWMM catchment were either adapted from the TEB model or derived from SWMM guidelines. For the suction head parameter (in mm)—the average soil capillary suction along the wetting front—used in the selected water infiltration method (i.e., the Green-Ampt approach), which is not included in the TEB model, we utilized the formula provided in SWMM's soil characteristics table. This calculation was based on soil permeability values obtained from the TEB model. Meteorological inputs for SWMM, like precipitation, are tailored from TEB's climate dataset to ensure consistency. Moreover,




to achieve a harmonized representation of evaporation processes between TEB and SWMM, the Penman-Monteith equation for potential evapotranspiration (PET) (Allen et al., 1998) is employed, leveraging climate forcing data and energy parameters computed within TEB. This integrated approach ensures a cohesive and accurate comparison between the hydrological behaviours modelled by TEB and SWMM.

The SUDS facility was modelled within SWMM bioretention module. This module necessitates the configuration of three

critical parameter groups for the SUDS: surface characteristics, soil properties, and the underground storage system. Parameter values are derived either directly from those determined for $E - SUDSa$ applications in TEB, encompassing reservoir and substrate features, or from SWMM-recommended values (Rossman and Huber, 2015) that have been refined to mirror the parameters used in TEB. This applies to the SUDS soil conductivity slope parameter—the average slope of the log(conductivity) versus soil moisture deficit curve (porosity minus moisture content, unitless)—which is not utilized in the

TEB model. For this parameter, we applied the formula proposed in SWMM's soil layer guidelines, which is based on the percentages of sand and clay in the soil. The LID was introduced into the catchment where the impervious and pervious surfaces are located.

Table 2 summarizes all the main parameters needed in TEB and SWMM for catchment and SUDS modelling.

**Table 2: Main parameters required for the modelling of $E - SUDSa$ type in TEB model and LID bioretention-type in SWMM model**

| TEB | | | | SWMM | | | |
|---|---|---|---|---|---|---|---|
| Description | | Quantity | Unity | Description | | Quantity | Unity |
| General | Specific | | | General | Specific | | |
| **Catchment parameters** | | | | | | | |
| | Garden | 0.196 | - | | Pervious | 1962.6 | m² |
| | Building | 0.500 | - | | Impervious | 7800.0 | m² |
| Land use fractions | Road | 0.280 | - | Sub catchment | | | |
| (mesh size = 1 ha) | $E - SUDSa$ | 0.024 | - | surfaces | LID | 238.4 | m² |
| | Impervious surface fraction | 0.78 | - | | Impervious surface percentage | 78.0 | % |
| Height of depression storage on impervious area | Building | 2 | mm | Height of depression storage on impervious area | --- | 2 | mm |
| | Road | 2 | mm | | | | |
| **$E - SUDSa$ /LID parameters** | | | | | | | |
| **Surface compartment** | | | | | | | |
| Maximum storage-infiltration (SI) reservoir height | $W_{SIMAX}$ | 100 | mm | Berm height | $D_1$ | 100 | mm |





| Substrate compartment | | | | | | | |
|---|---|---|---|---|---|---|---|
| Thickness | $THK_{subs}$ | 600 | mm | Thickness | $D_2$ | 600 | mm |
| Porosity | $\omega_{sat}$ | 0.43 | $m^3 \cdot m^{-3}$ | Porosity | $\theta_{sat}$ | 0.43 | $m^3 \cdot m^{-3}$ |
| Field capacity | $\omega_{FC}$ | 0.26 | $m^3 \cdot m^{-3}$ | Field capacity | $\theta_{FC}$ | 0.26 | $m^3 \cdot m^{-3}$ |
| Wilting point | $\omega_{WP}$ | 0.10 | $m^3 \cdot m^{-3}$ | Wilting point | $\theta_{WP}$ | 0.10 | $m^3 \cdot m^{-3}$ |
| Conductivity | $k_{sat}$ | 24.9 | $mm \cdot h^{-1}$ | Conductivity | $k_{sat}$ | 24.9 | $mm \cdot h^{-1}$ |
| Initial water content | $\omega_{ini}$ | 0.18 | $m^3 \cdot m^{-3}$ | Initial water content | $\theta_{ini}$ | 0.18 | $m^3 \cdot m^{-3}$ |
| Soil matrix potential at saturation | $\psi_{sat}$ | -210.0 | mm | Suction head | $\Psi_2$ | 82.76 | mm |
| Empirical parameter for the shape of the soil water retention curve | $b$ | 0.42 | - | Initial deficit | $\theta_{sat} - \theta_{ini}$ | 0.25 | $m^3 \cdot m^{-3}$ |
| | | | | Conductivity slope | $HCO$ | 31.28 | - |
| Storage compartment | | | | | | | |
| Maximum storage-exfiltration (SR) reservoir height | $h_{SEMAX}$ | 400 | mm | Thickness | $D_3$ | 400 | mm |
| Void ratio | $\phi_{SE}$ | 0.4 | - | Void ratio | $\phi_3$ | 0.4 | - |
| Exfiltration rate | $k_{sat}$ | 24.9 | $mm \cdot h^{-1}$ | Exfiltration rate | $k_{3s}$ | 24.9 | $mm \cdot h^{-1}$ |

### 4.2. Scenario 2: Combining multiple SUDS of the same type with different sizing configurations within the $E - SUDSa$.

In this scenario a combination of three SUDS facilities of type $R1C3$ (Figure 1) are simulated in two different manners:

- **Aggregated approach:** Three separate simulations are performed, each with a different $R1C3$ type configuration within a single mesh, and modelled hydrological fluxes are added together.
- **Equivalent $E - SUDSa$ approach:** A single simulation is conducted with a mesh area equivalent to the combined area of the three meshes in the aggregated approach. This mesh contains a single equivalent $R1C3$ configuration, whose characteristics are the average of the three $R1C3$ configurations from the aggregated approach.




The primary objective of this scenario is to undertake a comparative analysis of the hydrological efficiency between the
aggregated $R1C3$ SUDS and the $E - SUDSa$. This analysis aims to evaluate the errors introduced by the simplifications
inherent in the equivalent approach.

### 4.2.1 Methodology of the aggregated approach

1. **Urban Mesh Configuration in TEB**: Three urban meshes were generated in TEB, each covering an area of 1 hectare
   (10,000 m²) and representing various urban land uses. Detailed configurations are provided in Table 3. These values
were selected to illustrate different levels of urbanization, ranging from predominantly green areas to balanced
   environments, and densely built-up areas dominated by impervious surfaces.

2. **SUDS Implementation in Each Mesh (Figure 4)**: Each mesh integrates a SUDS facility of type $R1C3$, adapted to
   efficiently manage runoff from impervious areas (via the OASIS tool, Sage et al. (2024)). The SUDS in each mesh
   feature distinct sizing for the surface, substrate, and underground storage compartments. The specific design
parameters for each SUDS configuration are detailed in Table 3.

### 4.2.2 Methodology of the equivalent approach (Figure 4):

The $E - SUDSa$ is conceptualized by integrating the spatial extents and functionalities of all key urban elements—buildings,
roads, gardens, and SUDS—across the three meshes into a unified 3-hectare composite mesh. The surface of the $E - SUDSa$
represents the sum of three individuals $R1C3$ SUDS areas.

The dimensional parameters of the $E - SUDSa$, including reservoir heights and substrate thicknesses, are determined through
weighted averages of each corresponding dimension across the three SUDS configurations, factoring in the SUDS's area within
each mesh ($S_{SUDS-M*}$). This is articulated through the equations:

$$\overline{W}_{SIMAX} = \frac{W_{SIMAX-M1} \cdot S_{SUDS-M1} + W_{SIMAX-M2} \cdot S_{SUDS-M2} + W_{SIMAX-M3} \cdot S_{SUDS-M3}}{S_{SUDS-M1} + S_{SUDS-M2} + S_{SUDS-M3}}, \quad (28)$$

$$\overline{THK}_{subs} = \frac{THK_{subs-M1} \cdot S_{SUDS-M1} + THK_{subs-M2} \cdot S_{SUDS-M2} + THK_{subs-M2} \cdot S_{SUDS-M3}}{S_{SUDS-M1} + S_{SUDS-M2} + S_{SUDS-M3}}, \quad (29)$$

$$\overline{W}_{SEMAX} = \frac{W_{SEMAX-M1} \cdot S_{SCM-M1} + W_{SEMAX-M2} \cdot S_{SCM-M2} + W_{SEMAX-M3} \cdot S_{SCM-M3}}{S_{SUDS-M1} + S_{SUDS-M2} + S_{SUDS-M3}}. \quad (30)$$

This simple methodology (Eq. ( 28 ) – ( 30 )) allows verification of whether the $E - SUDSa$ design effectively represents the
collective characteristics of the individual SUDS configurations, facilitating a nuanced evaluation of integrated SUDS
strategies for enhancing urban hydrological management.

The application of the designated equations ( 28 ) – ( 30 ) yielded specific outcomes: a storage-infiltration reservoir height
($\overline{W}_{SIMAX}$) of 141 mm, a substrate thickness ($\overline{THK}_{subs}$) of 0.37 m and a storage-exfiltration height ($\overline{W}_{SEMAX}$) of 268 mm.
However, these calculated parameters for the substrate thickness and the storage-exfiltration reservoir height do not conform
to the specifications of the ISBA-DF soil depth grid (in meters) as employed in the TEB model: **[0.001, 0.01, 0.05, 0.10, 0.15,**
**0.20, 0.30, 0.60, 1.00, 1.50, 2.00, 3.00].** Consequently, modifications to these parameters are needed to achieve compatibility



within the framework of the TEB model. To address this discrepancy, two distinct configurations of the $E - SUDSa$ were devised and examined:

- **Configuration 1 ($E - SUDSa1$):** Maintains the calculated storage-infiltration height of 141 mm. The model adjusts the substrate layer to 0.60 m and sets the storage-exfiltration reservoir height to 400 mm.
- **Configuration 2 ($E - SUDSa2$):** Also adopts the storage-infiltration height of 141 mm. This version employs a substrate thickness of 0.30 m and a storage-exfiltration height of 300 mm.

The hydroclimatic data, soil characteristics, and vegetation type are the same as those used for Scenario 1 in the TEB model (based on the Pin Sec catchment, Table 2). Unlike scenario 1, for this scenario, since only the TEB model is used, lateral transfers of water flux between the different compartments of the TEB mesh are activated. Table 3 summarizes all the main parameters needed in TEB for Scenario 2 modelling.





**Table 3: Recapitulation of the parameters required for the development of scenario 2 in the TEB model**

| Description | | | Mesh 1 | Mesh 2 | Mesh 3 | $E-SUDSa1$ | $E-SUDSa2$ |
|---|---|---|---|---|---|---|---|
| General | Specific | Unity | Quantity | Quantity | Quantity | Quantity | Quantity |
| **Land use characteristics** | | | | | | | |
| Land use fractions | Garden | - | 0.591 | 0.407 | 0.105 | 0.368 | 0.368 |
| | Building | | 0.253 | 0.377 | 0.528 | 0.386 | 0.386 |
| | Road | | 0.138 | 0.198 | 0.347 | 0.228 | 0.228 |
| | SUDS | | 0.018 | 0.018 | 0.019 | 0.018 | 0.018 |
| | Sum | | 1.000 | 1.000 | 1.000 | 1.000 | 1.000 |
| Land use surfaces | Garden | m² | 5905.8 | 4066.2 | 1053.1 | 11025.2 | 11025.2 |
| | Building | | 2532.5 | 3770.4 | 5284.7 | 11587.6 | 11587.6 |
| | Road | | 1379.1 | 1983.8 | 3474.5 | 6837.4 | 6837.4 |
| | SUDS | | 182.6 | 179.6 | 187.7 | 549.8 | 549.8 |
| | Sum | | 10000.0 | 10000.0 | 10000.0 | 30000.0 | 30000.0 |
| **Surface mesh characteristics** | | | | | | | |
| Height of depression storage on impervious area | --- | mm | | | 2.0 | | |
| SUDS maximum storage-infiltration (SI) reservoir height | $W_{SIMAX}$ | | 50.0 | 120.0 | 250.0 | 141.0 | 141.0 |
| **Soil mesh characteristics** | | | | | | | |
| Total number of soil layers | --- | | | | 12.0 | | |
| Total soil depth | --- | m | | | 3.0 | | |
| Soil texture | Clay | % | | | 8.0 | | |
| | Silt | | | | 41.0 | | |
| | Sand | | | | 51.0 | | |
| Porosity | $\omega_{sat}$ | m³·m⁻³ | | | 0.43 | | |
| Field capacity | $\omega_{FC}$ | | | | 0.26 | | |
| Wilting point | $\omega_{WP}$ | | | | 0.10 | | |
| Initial soil water content | $\omega_{ini}$ | | | | 0.18 | | |
| Empirical parameter for the shape of the soil water retention curve | $b$ | - | | | 0.42 | | |
| Soil matrix potential at saturation | $\psi_{sat}$ | m | | | -0.21 | | |
| Permeability at saturation | $k_{sat}$ | m·s⁻¹ | | | 6.92E-06 | | |
| **SUDS underground structure characteristics** | | | | | | | |
| Number of soil layers representing the SUDS substrate (soil before the underground storage layer) | --- | - | 6.0 | 7.0 | 8.0 | 8.0 | 7.0 |
| Substrate depth (before underground storage layer) | $THK_{subs}$ | m | 0.2 | 0.3 | 0.6 | 0.6 | 0.3 |
| Maximum storage-exfiltration (SE) reservoir height | $W_{SEMAX}$ | mm | 100.0 | 300.0 | 400.0 | 400.0 | 300.0 |
| Depth of substrate + underground storage layer | --- | m | 0.3 | 0.6 | 1.0 | 1.0 | 0.6 |
| Width of the underground storage layer | $b_{SE}$ | m | 13.5 | 13.4 | 13.7 | 23.4 | 23.4 |
| Porosity of the storage layer (SE reservoir) | $\phi_{SE}$ | - | | | 0.4 | | |





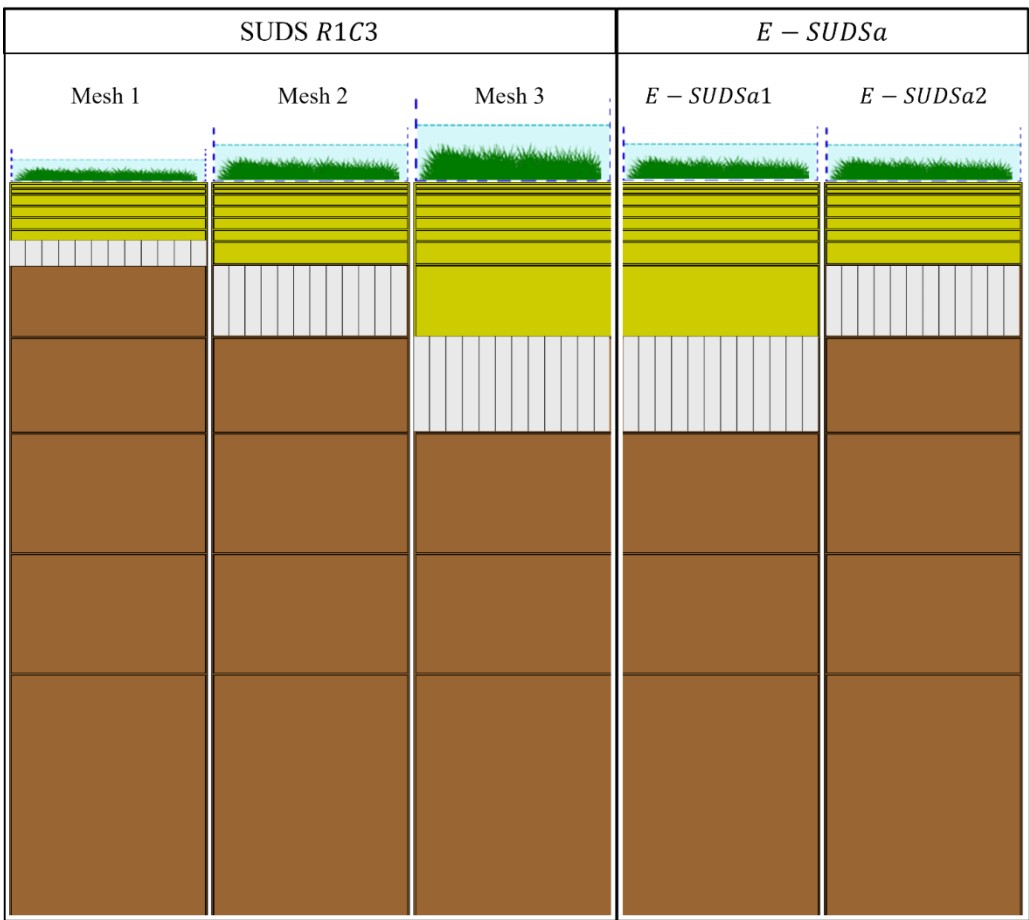

**Figure 4: Conceptualisation and vertical alignment of SUDS $R1C3$ and $E - SUDSa$ configurations within the TEB Model's soil grid for Scenario 2. Here, the substrate layers are depicted in yellow, indicating their specific positioning, while the underlying natural soil beneath the SUDS/$E - SUDSa$ configurations is shown in brown.**

**4.3. Comparative evaluation of hydrological processes in the scenarios**

In the comparative analysis between the SWMM and TEB models, we examine the different hydrological processes within the SUDS. These processes encompass inflow (rainfall and runoff), infiltration, evapotranspiration, percolation, exfiltration, overflow, and the storage dynamics within the SI reservoir. Another aspect of this comparison involves assessing the water content within SWMM's LID module—a singular layer substrate—against the weighted average water content across the $E - SUDSa$'s eight substrate layers, with weighting by layer thickness. Moreover, the analysis extends to comparing the total water balance over the study period for each SUDS model.

The water balance error for SWMM/TEB models is defined as follows:

$$e_{WB-*}(\%) = \frac{\sum_{t=1}^{nb} Q_{in-*,t} - \left[\sum_{t=1}^{nb} Exf_{\cdot *,t} + \sum_{t=1}^{nb} Ev_{\cdot *,t} + \sum_{t=1}^{nb} q_{*,t} + \Delta S_{*,nb}\right]}{\sum_{t=1}^{nb} Q_{in-*,t}} \cdot 100,$$ ( 31 )





where, $e_{WB-*}$ represents the percentage errors in the water balance for the respective model (*), calculated over the total number ($nb$) of time steps. Here $Q_{in-*,t}$ is the inflow (mm) at each time step $t$, $Ev._{*,t}$ evapotranspiration (mm), $Exf._{*,t}$ the water exfiltration (mm), $q_{*,t}$ the outflow (mm) and $\Delta S_{*,nb}$ the net change in water storage within the SUDS structure, accounting for the surface, substrate, and underground storage compartments (mm), at the final interval $nb$.

For scenario 2, similar to scenario 1, the performance of hydrological processes, resulting from the aggregation of the 3 SUDS and the 2 $E-SUDSa$ configurations, has been compared. The eight processes to be analysed are as follows: inflow resulting from rainfall and runoff originating from impervious surfaces connected to the SUDS system, water infiltration into the substrate of the SUDS system, evapotranspiration dynamics, potential overflow occurrences, substrate percolation into the SE reservoir, deep drainage from the final soil layer within the soil column (at a depth of 3 meters), and water storage within the

SUDS structure (including SI reservoir, substrate soil layers, SR reservoir) as well as in the soil layers beneath the SUDS system (Figure 4).

For the analysis of these processes, two commonly used performance indices in the field of hydrology have also been applied (N. Moriasi et al., 2007): the Nash-Sutcliffe efficiency coefficient (NSE) and the percent bias (PBIAS), the formulas for which are as follows:

$$NSE = \frac{\sum_{t=1}^{nb}\left( Q_{sim,t} - Q_{ref,t}\right)^2}{\sum_{t=1}^{nb}\left( Q_{ref,t} - \overline{Q_{ref,t}}\right)^2}, \qquad (32)$$

$$PBIAS = \frac{\sum_{t=1}^{nb}\left( Q_{ref,t} - Q_{sim,t}\right)}{\sum_{t=1}^{nb} Q_{ref,t}} \cdot 100, \qquad (33)$$

with $nb$ the number of time steps, $Q_{sim,t}$ and $Q_{ref,t}$ respectively simulated and reference hydrological fluxes at time step $t$ (L·s⁻¹). $\overline{Q_{obs,t}}$ is the arithmetic mean of reference hydrological fluxes (L·s⁻¹).

For the application of these two indices, the processes generated by the aggregation of the 3 SUDSs were assumed as reference data ($Q_{ref,t}$), while the simulated data ($Q_{sim,t}$) were produced separately by the $E-SUDSa1$ and $E-SUDSa2$ configurations.

In addition to the performance of hydrological processes, an analysis similar to that in scenario 1 was conducted on the water balance for both the three aggregated SUDS and the two configurations of the $E-SUDSa$ type.

In contrast to Scenario 1, the computation of the water balance in Scenario 2 integrates additional hydrological components: deep soil water drainage ($Ddr$) from the bottommost natural soil layer with SUDS compartment in the TEB mesh and lateral soil water transfer ($Lf$). These variables augment the existing model framework, with deep drainage supplanting the role of

exfiltration as delineated in Eq.( 31 ), and lateral transfer flux reintroduced after being previously omitted to ensure comparability with the SWMM model. Thus, the recalibrated water balance error is articulated as:

$$e_{WB-*}(\%) = \frac{\sum_{t=1}^{nb} Q_{in-*,t} - \left[\sum_{t=1}^{nb} Ddr._{*,t} + \sum_{t=1}^{nb} Lf._{*,t} + \sum_{t=1}^{nb} Ev._{*,t} + \sum_{t=1}^{nb} q_{*,t} + \Delta S_{*,nb}\right]}{\sum_{t=1}^{nb} Q_{in-*,t}} \cdot 100. \qquad (34)$$

In this equation, the asterisk (*) represents the collective entity of the three SUDS types or the individual $E-SUDSa$ configurations. The term $e_{WB-*}$ represents the percentage error in the water balance, $nb$ indicates the total number of discrete



time steps, $Q_{in-*,t}$ is the inflow (m³) at each interval $t$, $Ev._{*,t}$ accounts for evapotranspiration (m³), $Ddr._{*,t}$ denotes the deep

drainage (m³), $q_{*,t}$ refers for overflow (m³), $Lf_{*,t}$ represents lateral soil water transfer (m³) and $\Delta S_{*,nb}$ represents the net change in water storage within the SUDS system (i.e., surface, substrate and underground storage compartment) as well as in the soil layers beneath SUDS system at the conclusion of the period $nb$ (m³).

## 4.4. $E-SUDSa$ hydrological evaluation results and discussion

### 4.4.1 Scenario 1

For scenario 1, the hydrological performance of the $E-SUDSa$ was evaluated against the bioretention LID-SWMM model, focusing on key SUDS processes (Figure 5). Scatter plots were used to illustrate the simulation relationships, with a 1:1 correspondence line (in red) representing identical results between the SWMM and TEB models and a best-fit line (in blue) representing precision as measured by the value of $R^2$ value.

The proximity of data points, especially for inflow and overflow to the red line underscores the $E-SUDSa$ model's high

accuracy in simulating inflow and overflow dynamics, paralleling the performance of the SWMM model

For percolation and exfiltration fluxes, as well as water storage in the surface and substrate, we observed a good correlation between the TEB and SWMM models. However, these simulations yielded by the two models show greater dispersion (Figure 5). The analysis of water storage within SUDS substrate highlighted a significant correlation between the models for soil water contents at or above field capacity ($\omega_{FC}$=0.28). Below this threshold, model congruence decreases, attributed to differences in

the initiation of percolation between the models. Specifically, the SWMM model initiates percolation from the substrate into underground storage at field capacity, while the TEB model applies a free- drainage boundary condition, allowing percolation to continue at a rate equal to the permeability of the lower soil layer. This discrepancy leads to variations in simulation results for the percolation and exfiltration processes. Surface water storage evaluation reveals that $E-SUDSa$ generally underestimates reservoir levels compared to LID-SWMM. This discrepancy is attributed to differences in the surface water

balance process calculation order in each model. In SWMM, overflow is considered after evapotranspiration and infiltration, whereas in the TEB model, overflow is calculated before these processes (Eq.( 1 )). This discrepancy may also be due to differences in the infiltration fluxes yielded by the two models.

The results for infiltration and evapotranspiration fluxes show significant differences between the two models (Figure 5). When examining infiltration performance, it is evident that the $E-SUDSa$ can infiltrate a greater volume of inflow compared to the

LID-SWMM model. This discrepancy is due to the different infiltration methods employed by each model. The $E-SUDSa$ employs the default infiltration method of ISBA, where infiltration is calculated based on the difference between the actual water content in the soil column and the maximum water content capacity of the soil (Albergel et al., 2018). In contrast, SWMM employs the Green-Ampt method, which considers the soil has maximum infiltration capacity . (Rossman and Huber, 2015). However, studies (Kale and Sahoo, 2011; Niazi et al., 2017) indicate that the Green-Ampt method may not adequately

address the complexity of urban soil heterogeneity, an area where the ISBA methodology aims to provide a more dynamic and



physically coherent framework (Vereecken et al., 2019). If the infiltration yielded by the TEB model is less significant than that of the SWMM model, it can be assumed that most of the water stored on the SUDS surface has been primarily evacuated through evapotranspiration processes.

For evapotranspiration, $E - SUDSa$ yields higher rates than the LID-SWMM model (Figure 5). The TEB model adopts a

detailed energy balance methodology that accounts for the intricate exchange of various energy fluxes, notably solar radiation and ambient thermal conditions (Stavropulos-Laffaille et al., 2021). Despite its holistic framework, the energy balance model's reliance on multiple interdependent parameters invoked criticisms highlighting possible rate overestimations. (Ouédraogo et al., 2023; Vera et al., 2018). The TEB model's evapotranspiration predictions have been found to overestimate the observed values, especially in seasons such as spring and autumn (Stavropulos-Laffaille et al., 2021). In contrast, the LID-SWMM

model employs a more reductive water balance technique based on predetermined potential evapotranspiration rates, commonly used by other hydrological models (Zhao et al., 2013). This simplified approach may not adequately capture the intricate dynamics influenced by climatic variability and the heterogeneity of urban areas (Hörnschemeyer et al., 2021; Ouédraogo et al., 2023), leading to deficiencies in modelling urban SUDS evapotranspiration (Hörnschemeyer et al., 2021, 2023).

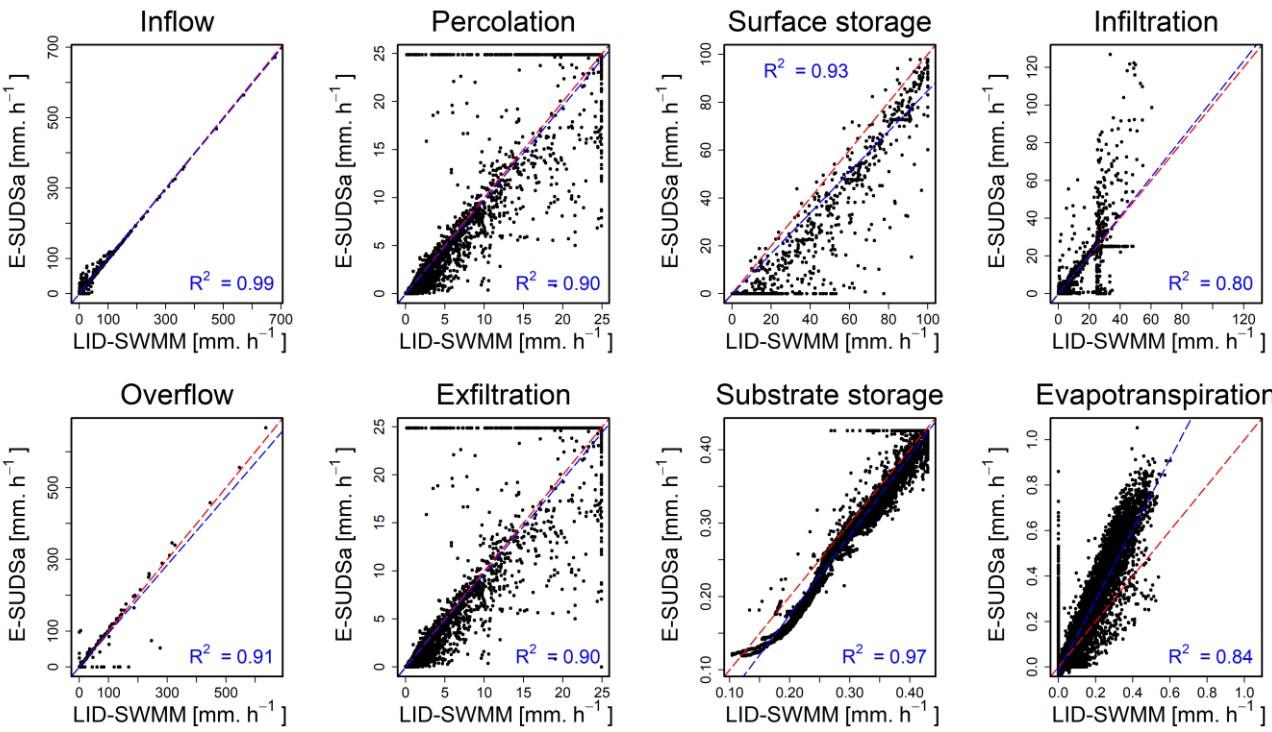


**Figure 5: Comparison of different SUDS hydrological processes between SWMM and TEB models from May 2010 to August 2012 (1-hour time step).**



Although both models have drawbacks regarding evapotranspiration, it is essential to underscore that the TEB model's approach is substantially more detailed, capturing a broader spectrum of hydro-energetic interactions. This level of detail is

particularly crucial for assessing the effectiveness of vegetative SUDS, which are significantly influenced by evapotranspiration dynamics.

Figure 6 and Table 4 present the results of the SUDS water balance from May 2010 to August 2012 yielded by SWMM and TEB models. The SUDS was designed to intercept 80% of total rainfall, thus limiting the overflow to a maximum of 20% of the total inflow ($Q_{in}$).

The inflow analysis indicates a minor discrepancy between the two models, with SWMM showing a slightly higher value (4.23e+04 mm) compared to TEB (4.15e+04 mm). This difference is attributed to the modelling approaches for runoff generation from impervious surfaces: SWMM utilizes a non-linear reservoir model, capturing the variable runoff response to rainfall events, while TEB employs a simpler reservoir overflow method, potentially underpredicting inflow during high-intensity rainfall periods.

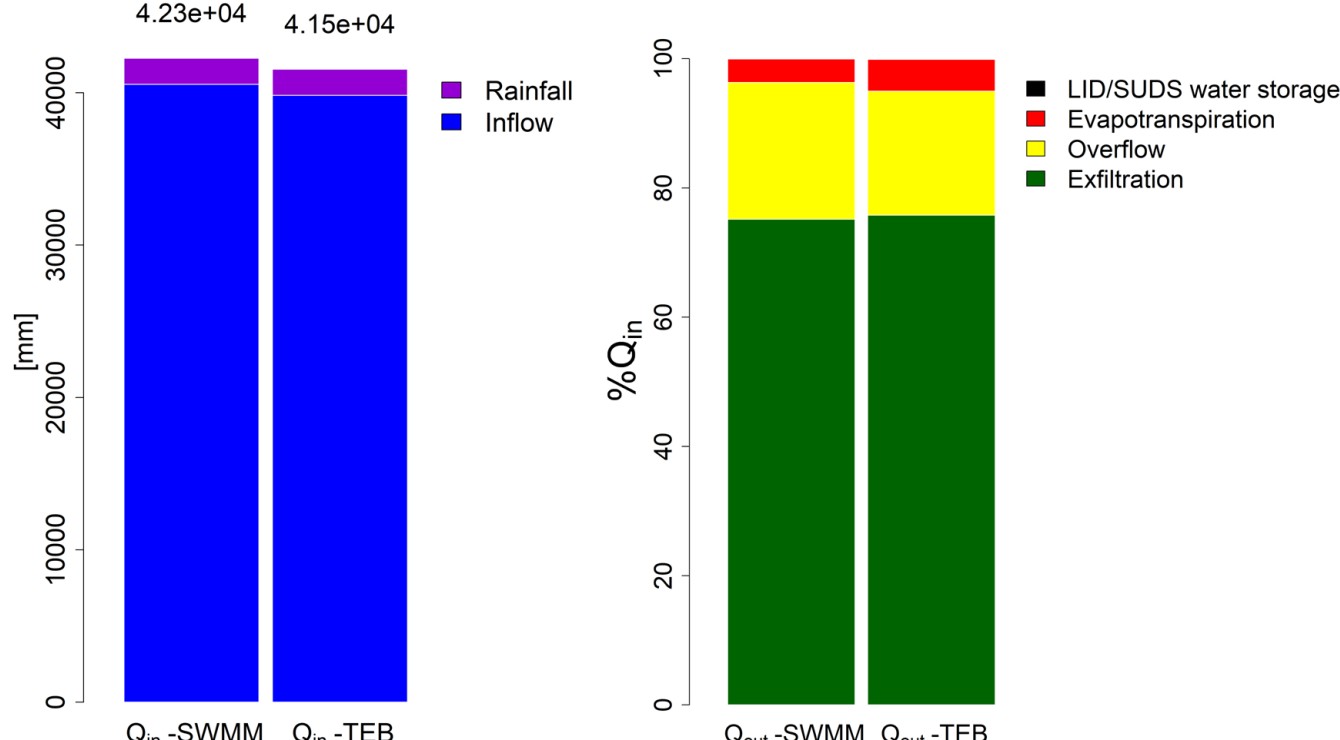


**Figure 6: Comparative water balance outcomes of SWMM and TEB SUDS models from May 2010 to August 2012. The left side of the graph displays bar plots of the inflows to SUDS (in mm), while the right side shows the outflows from SUDS (as a percentage of inflows - $Q_{in}$)**

The TEB model exhibited a higher rate of evapotranspiration and greater water storage changes within the SUDS, reflective

of its sensitivity to water retention and plant water use. However, these processes account for only a small fraction of the



overall water balance. Conversely, SWMM's higher overflow may imply a less detailed representation of these processes. Notably, both models showed exfiltration as the largest output component, suggesting that soil permeability and water seepage into the subsurface are robustly represented. The analysis revealed both models' precision (Eq. ( 31 )) in simulating the water balance, with a short error for SWMM and effectively zero for TEB.

Future research should aim to calibrate $E - SUDSa$ against observed data, ensuring that the predicted hydrological behaviours closely match real conditions.

**Table 4: Hydrological variables used in the computation of SUDS water balance for the SWMM and TEB models**

| Variable | Parameter [$P$] | SWMM | | TEB | |
|---|---|---|---|---|---|
| | | Quantity [mm] | $P/\sum Q_{in}$ [%] | Quantity [mm] | $P/\sum Q_{in}$ [%] |
| $Q_{in}$ | Rainfall | 1.71e+03 | 4.05 | 1.71e+03 | 4.12 |
| | Inflow | 4.05e+04 | 95.95 | 3.98e+04 | 95.88 |
| | $\sum Q_{in}$ | 4.23e+04 | 100.00 | 4.15e+04 | 100.00 |
| $Q_{out}$ | LID/$E - SUDSa$ water storage ($\Delta S$) | 0.29e-02 | 0.07 | 0.47e-02 | 0.11 |
| | Evapotranspiration ($Ev$) | 1.54e+03 | 3.64 | 2.04e+02 | 4.91 |
| | Overflow ($q$) | 8.94e+03 | 21.15 | 7.98e+03 | 19.22 |
| | Exfiltration ($Exf$) | 3.18e+04 | 75.14 | 3.15e+04 | 75.76 |
| | $\sum Q_{out}$ | 4.23e+04 | 100.00 | 4.15e+04 | 100.00 |
| $e_{WB-*}[\%] = \dfrac{(\sum Q_{in} - \sum Q_{out})}{\sum Q_{in}}$ | | -1,96e-03 | | 5.33e-11 | |

In conclusion, the analysis of scenario 1 demonstrates that the conceptualization of $E - SUDSa$ is accurate for the selected configuration. However, to fully validate the $E - SUDSa$ module, this analysis should be extended to include a variety of

soil/substrate textures and reservoir sizes.

**4.4.2 Scenario 2**

Figure 7 presents a comparison of the dynamics in hydrological processes yielded by the aggregation of the 3 different SUDS (SUM-SUDS) and the two configurations proposed by $E - SUDSa$. Generally, the simulations compared between SUM-SUDS and $E - SUDSa$ (both configurations). Both approaches produce similar results in the processes of inflow, overflow

and SUDS water storage. While there is a good correlation between SUM-SUDS and $E - SUDSa$ in other analysed processes, some dispersion can be observed, particularly in the percolation and exfiltration processes.

In the case of infiltration and evapotranspiration, depending on the substrate thickness, the simulations either overestimate ($E - SUDSa1$) or underestimate ($E - SUDSa2$) the results compared to SUM-SUDS. For percolation and exfiltration, in addition to soil thickness, the influence of infiltration and evapotranspiration processes on the dispersion of the compared

simulations can also be observed.





Among the hydrological processes analysed in Figure 7, $E - SUDSa2$ demonstrates better performance than $E - SUDSa1$.

**Figure 7: Comparison of SUDS hydrological processes of the three aggregated SUDS (SUM-SUDS) against proposed $E - SUDSa1$ (orange dots) and $E - SUDSa2$ (violet dots) configurations from May 2010 to August 2012 (1-hour time step).**





Table 5 presents the NSE and PBIAS values for both $E - SUDSa$ configurations across the evaluated hydrological processes against SUM-SUDS. The NSE values indicate a good overall dynamic representation of processes, although there is greater difficulty with exfiltration and percolation (NSE = 0.9 or 0.93, depending on the configuration). For PBIAS, the bias is generally low (<5% absolute) for the evaluated hydrological processes, except for evapotranspiration and overflow, where $E - SUDSa$1 shows absolute biases exceeding 10%. In contrast, $E - SUDSa$2 exhibits only a 5.8% bias for overflow, with

all other processes showing biases between -1.10% and 1.6%. Based on both NSE and PBIAS scores, $E - SUDSa$2 demonstrates better performance than $E - SUDSa$1.

Table 5: NSE and PBIAS values obtained for $E - SUDSa$1 and $E - SUDSa$2

| SUDS hydrological processes | $E - SUDSa$1 | | $E - SUDSa$2 | |
|---|---|---|---|---|
| | NSE [-] | PBIAS [%] | NSE [-] | PBIAS [%] |
| Inflow | 1.00 | 1.60 | 1.00 | 1.60 |
| Evapotranspiration | 0.92 | 17.50 | 0.99 | 0.80 |
| Infiltration | 0.86 | 4.20 | 0.92 | 1.40 |
| Overflow | 0.95 | -12.00 | 0.95 | 5.80 |
| SUDS water storage | 0.97 | -6.20 | 0.99 | -1.10 |
| Percolation | 0.90 | 2.20 | 0.93 | 1.30 |
| Exfiltration | 0.90 | 2.20 | 0.93 | 1.30 |
| Deep Drainage | 0.94 | 5.40 | 0.96 | 1.20 |

The results of the water balance for the aggregated SUDS models (SUM-SUDS) and the two proposed configurations of $E -$

$SUDSa$, covering the period from May 2010 to August 2012, are detailed in Figure 8 and Table 6. The negligible water balance errors ($e_{WB-*}$) which are practically zero for the three configurations under study, underscore the high accuracy of the water balance calculations, thereby validating the precision of the TEB model. Inflow inputs maintain uniformity across all configurations (around 96%). Regarding the processes produced by the SUDS, the overall results are quite similar between SUM-SUDS and the two $E - SUDSa$ configurations. The main differences are observed in $E - SUDSa$1, particularly in the

deep soil water drainage and overflow processes, where there is an approximate 2.3% difference compared to SUM-SUDS and $E - SUDSa$2.

The evaluation conducted in Scenario 2 suggests that the equivalence approach is suitable for representing the aggregation of SUDS (SUM-SUDS) with similar structure and hydrologic processes into a single entity ($E - SUDSa$). The evaluation corroborates the adequacy of the parameterization approach used in the $E - SUDSa$ reservoirs. Of the two configurations

tested, $E - SUDSa$2 demonstrates better performance than $E - SUDSa$1. The assessment of temporal dynamics between SUM-SUDS and $E - SUDSa$ (Figure 7) shows that, for almost all hydrological processes compared, the performances are similar. However, differences are observed in percolation and exfiltration during certain events. As indicated in the water





balance, exfiltration is the most significant process produced by these SUDS, generally having the greatest influence on SUDS

of this configuration type (i.e., $R1C3$) (Grey et al., 2018).

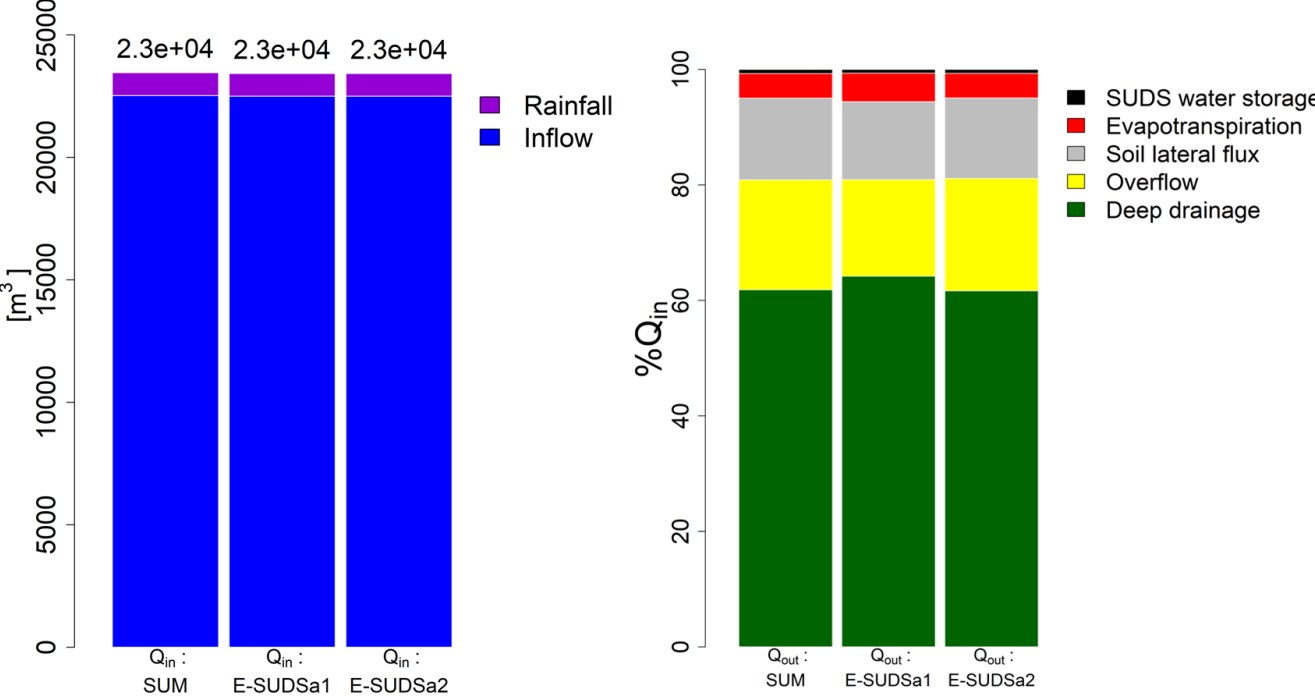


**Figure 8: Comparative water balance outcomes of the three aggregated SUDSs (SUM) and the two $E - SUDSa$ configurations from May 2010 to August 2012. The left side of the graph displays bar plots of the inflows to SUDS (in m³), while the right side shows the outflows from SUDS (as a percentage of inflows - $Q_{in}$)**



**Table 6: Hydrological variables used in the computation of SUDS water balance for the three aggregated SUDS (SUM-SUDS) and the two $E - SUDSa$ configurations.**

| Variable | Parameter [P] | SUM-SUDS | | $E - SUDSa1$ | | $E - SUDSa2$ | |
|---|---|---|---|---|---|---|---|
| | | Quantity [m³] | $P/\sum Q_{in}$ [%] | Quantity [m³] | $P/\sum Q_{in}$ [%] | Quantity [m³] | $P/\sum Q_{in}$ [%] |
| $Q_{in}$ | Rainfall | 9.00e+02 | 4.02 | 9.00e+02 | 3.95 | 9.00e+02 | 3.95 |
| | Inflow | 2.25e+04 | 95.98 | 2.25e+04 | 96.05 | 2.25e+04 | 96.05 |
| | $\sum Q_{in}$ | 2.346e+04 | 100.00 | 2.342e+04 | 100.00 | 2.342e+04 | 100.00 |
| $Q_{out}$ | SUDS water storage ($\Delta S$) | 1.645e+02 | 0.70 | 1.522e+02 | 0.65 | 1.591e+02 | 0.68 |
| | Evapotranspiration ($Ev$) | 1.003e+03 | 4.27 | 1.159e+03 | 4.95 | 9.944e+02 | 4.25 |
| | Soil water lateral flux ($Lf$) | 3.320e+03 | 14.15 | 3.160e+03 | 13.49 | 3.268e+03 | 13.95 |
| | Overflow ($q$) | 4.466e+03 | 19.03 | 3.913e+03 | 16.71 | 4.561e+03 | 19.47 |
| | Deep soil water drainage ($Dd$) | 1.451e+04 | 61.84 | 1.504e+04 | 64.21 | 1.444e+04 | 61.65 |
| | $\sum Q_{out}$ | 2.346e+04 | 100.00 | 2.342e+04 | 100.00 | 2.342e+04 | 100.00 |
| $e_{WB-*}[\%] = \left.(\sum Q_{in} - \sum Q_{out})\middle/\sum Q_{in}\right. [\%] =$ | | 2.64e-13 | | -4.97e-13 | | 1.12e-15 | |

## 5 Conclusions

Sustainable Drainage Systems (SUDS) are critical elements in urban water management strategies, offering multifaceted benefits beyond stormwater control. Their comprehensive evaluation is essential, especially given the pressing challenges posed by global warming and rapid urban expansion. To fully capture the advantages of SUDS at the urban scale, integrating them into hydro-climatic models is necessary. The TEB model, renowned for its capabilities in hydro-climatic simulation at the urban level, has been adapted to develop an innovative SUDS module. This adaptation aligns with the $E - SUDS$ framework, which consolidates diverse SUDS into a unified entity based on shared hydrological processes and the unique parameters of the TEB model. This development represents a significant advancement in urban hydro-energetic modelling.

A first validation of the TEB-SUDS module was methodically conducted through two distinct modelling scenarios. The first scenario assessed the ability of one of the five $E - SUDS$ types ($E - SUDSa$) in the TEB model to replicate the hydrological dynamics of the LID bioretention module in the SWMM model. The results indicated that the selected configuration of $E - SUDSa$ was effective for most evaluated hydrological processes. The second scenario focused on evaluating the overall concept of the equivalent approach, based on the evaluated $E - SUDSa$. The results of this scenario demonstrate the ability of the equivalent SUDS approach to model various hydrological processes generated by the aggregation of SUDS of a same type, if they are adequately parameterised.

Future research will first focus on the evaluation and validation of each SUDS that forms part of the five proposed $E - SUDS$ (Table 1). For this purpose, the methodology described in Scenario 1 will be applied with suitable reference models.



An important aspect to explore involves low-permeability soils beneath the $E - SUDS$. This will allow for a detailed study of the functioning of the underground storage compartment under these boundary conditions. In this study, this scenario could

not be evaluated because the high permeability of the soil did not allow water to be retained in the compartment. In this context, it would be relevant to test, within $E - SUDSa$, a substrate and an underlying soil column with varying hydraulic parameters to assess their differences in terms of flow dynamics and water retention. It is worth noting that, to our knowledge, TEB is the only model among those incorporating a SUDS module that can perform such modelling. Additionally, it is crucial to evaluate sealed SUDS, particularly in $E - SUDSa$ and $E - SUDSb$, to analyze the hydrological effects induced by this boundary

condition in the substrate layer.

Applying the methodology developed in Scenario 2 to other $E - SUDS$ is also necessary. Unlike the current scenario, which tests a single type of SUDS, future research will prioritize the use of various combinations of SUDS types integrated into the $E - SUDS$. This approach will allow us to test whether the coupling hypotheses between different types of SUDS assembled in the $E - SUDS$ are correct.

Another key aspect to address concerns scenarios involving multiple $E - SUDS$ within the same mesh. In the developed module, each $E - SUDS$ currently receives a percentage of the runoff generated by impermeable surfaces, with this percentage defined by the user as input data. Each $E - SUDS$ then independently manages the received water, and any potential overflows are directed to the garden compartment of the TEB mesh. However, the module does not yet account for cases where the overflow produced by one $E - SUDS$ could become inflow for another $E - SUDS$, nor for regulated outflows produced by the

$E - SUDS$. A future study will focus on developing a detailed framework for interconnecting flows between different $E - SUDS$ within the same mesh to optimize stormwater management.

This article focuses on the functionality of the SUDS module applied to a single TEB mesh. However, to work at an urban scale, a grid of meshes must be considered. This will require the creation of scenarios where different $E - SUDS$ are implemented across multiple meshes, enabling an evaluation of the cumulative effects of SUDS utilization for stormwater

management at the urban scale.

To realize these perspectives, it would be ideal to have observed data from constructed SUDS on the field. In the absence of such data, the option of continuing to use existing hydrological models for validating the proposed perspectives will be considered.

Finally, this article does not address the energy and radiative processes developed for the $E - SUDS$. This omission is because,

to our knowledge, no hydrological model with an integrated SUDS module simulates such processes. To evaluate whether these processes are accurately represented in our SUDS module, acquiring specific observed data will be essential.





**Appendix A: New configuration of urban compartments of the TEB mesh**

**Figure A.1: Five $E - SUDS$ types represented with their respective reservoirs.**





**Appendix B: Conceptualisation of the other $E - SUDS$**



**Figure B.1: Conceptualisation of the hydrological functioning of the $E - SUDSb$ type**







**Figure B.2: Conceptualisation of the hydrological functioning of the $E - SUDSc$ type**





**Figure B.3: Conceptualisation of the hydrological functioning of the $E - SUDSd$ type**






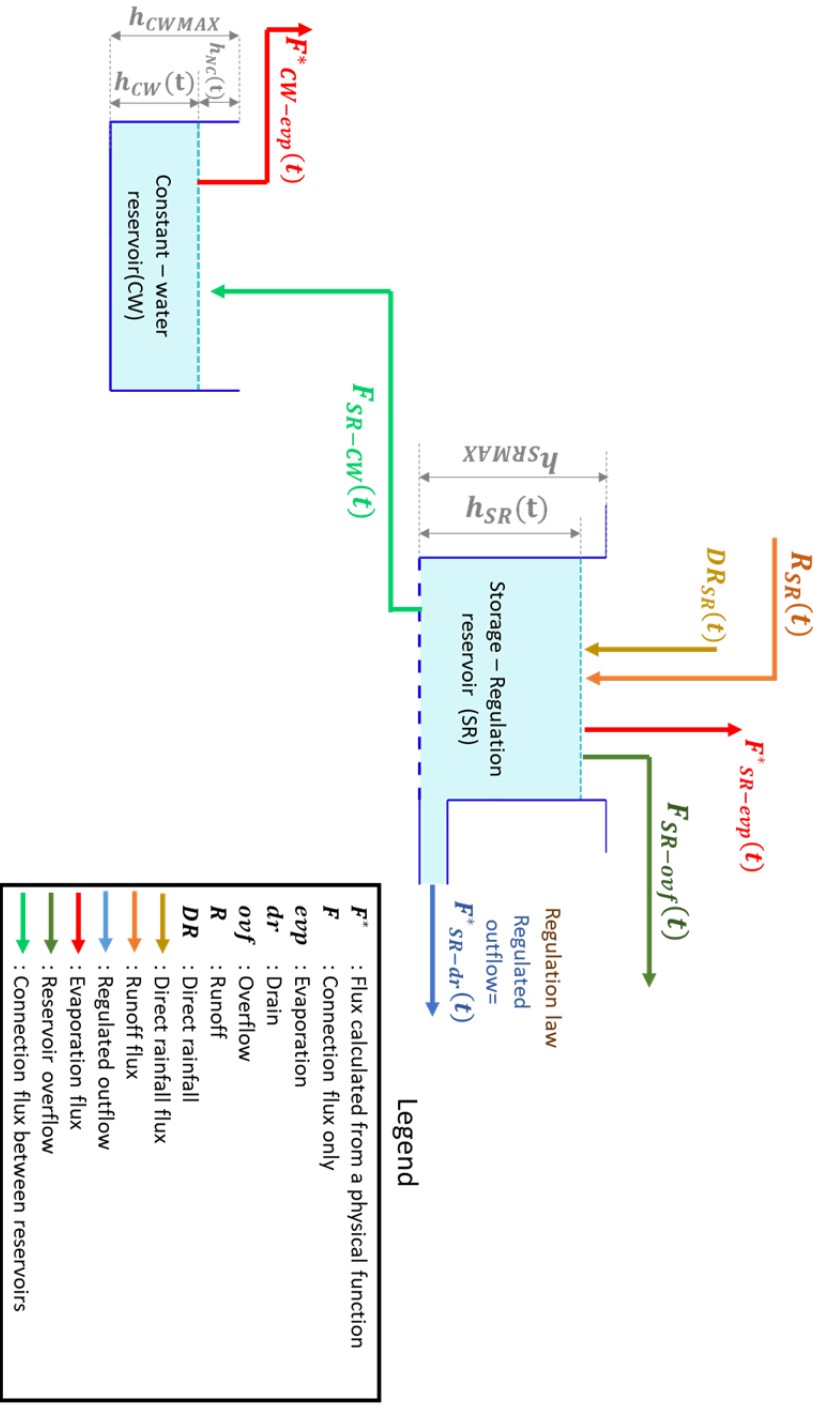

**Figure B.4: Conceptualisation of the hydrological functioning of the $E - SUDSe$ type**




**Appendix C: Definition of the $E - SUDS$ surface water evaporation formula**

In the ISBA-DF model, surface water evaporation occurs when surface overflow is generated due to the saturation of the soil
column in the vegetation (garden) fraction of the TEB mesh. This process is calculated using the following formula:

$$F_{evp}(t) = \frac{p_{ff} \cdot \rho_a \cdot C_H \cdot V_a \cdot [q_{sat}(T_s) - q_a]}{\Delta t},$$ (35)

where $p_{ff}$ (-) is the fraction $[0 - 1]$ of the surface overflow in the TEB mesh, calculated using the CTRIP model (Decharme
et al., 2012) coupled with the ISBA-DF model.

This equation forms the basis for our evaporation formula. In our case, we assume that $p_{ff}$ can be represented as the ratio $[0
- 1]$ of the volume of water stored in the surface reservoir ($V_*$)to the total storage capacity of the surface reservoir ($V_{*MAX}$).
Therefore, it can be expressed as:

$$F_{evp}(t) = \frac{\left(V_* / V_{*MAX}\right) \cdot \rho_a \cdot C_H \cdot V_a \cdot [q_{sat}(T_s) - q_a]}{\Delta t}.$$ (36)

$V_*$ and $V_{*MAX}$ can be expressed as the product of water levels ($W_*$ and $W_{*MAX}$) and the areas ($S_*$ and $S_{*MAX}$) occupied by this
water levels. In our case $S_* = S_{*MAX}$, allowing the water surface evaporation formula to be simplified as follows:

$$F_{evp}(t) = \frac{\left(W_* / W_{*MAX}\right) \cdot \rho_a \cdot C_H \cdot V_a \cdot [q_{sat}(T_s) - q_a]}{\Delta t}.$$ (37)

In addition, to consider the effects of snow on the SUDS surface, the same criterion used for the ponding surfaces of the
impervious areas of the TEB mesh (i.e., building and road) has been adapted (Masson,2000). This involves raising the ratio of
the water level in the reservoir to its maximum level to the power of 2/3. Consequently, the equation for water evaporation at
the SUDS surface is as follows:

$$F_{evp}(t) = \frac{\left(W_* / W_{*MAX}\right)^{2/3} \cdot \rho_a \cdot C_H \cdot V_a \cdot [q_{sat}(T_s) - q_a]}{\Delta t}.$$ (38)

As indicated in the main text, the surface of the $E - SUDSa$ and $E - SUDSb$ can be represented as a combination of SI and
SR reservoirs. In this configuration, when rainwater is stored in both reservoirs, only the water level of the SR reservoir is
taken for the evaporation calculation. Another case may be that the $E - SUDS$ is composed of SUDS that have only SI
reservoir on the surface (Example: SUDS R1C3 in the $E - SUDSa$) and SUDS that have a combination of SI and SR reservoirs
(Example: SUDS R2C5 in the $E - SUDSa$). In the case that both reservoirs contain water, the evaporation process is
considered for both reservoirs (SI and SR). However, the evaporation flux of the SI reservoir should only account for the
SUDS containing solely this reservoir. To ensure this, the fraction of SUDS with only an SI reservoir on the surface ($f_{SI}$, Eq.
( 2 )) is applied to the evaporation flux.

**Code and data availability**

The exact version of SURFEX v9.0, including the TEB model and the TEB-SUDS module (implemented here for the $E -
SUDSa$ type), used to generate the results of this study is archived on Zenodo and publicly available under the GPLv3 license



(https://doi.org/10.5281/zenodo.17144712) (Tunqui Neira et al., 2025). The repository also contains the input data required to
run the simulations and the output data used for their evaluation.

This study also makes use of the U.S. EPA Storm Water Management Model (SWMM), version 5.2.4. The SWMM source
code and binaries are freely available from the official U.S. EPA distribution (https://www.epa.gov/water-research/storm-
water-management-model-swmm). The input and output files associated with the SWMM simulations performed in this study
have likewise been archived in the same Zenodo repository (https://doi.org/10.5281/zenodo.17144712) (Tunqui Neira et al.,
720    2025)

**Author contributions**

JMTN led the model development, designed and evaluated the simulation scenarios, and wrote the manuscript. KC, MCG, and
GC supervised the project, provided expertise in coding and hydrological modelling, and contributed to the review and revision
of the manuscript.

**Competing interests**

The authors declare that they have no conflict of interest.

**Acknowledgements**

This research was carried out under the OPUR research program (https://www.leesu.fr/opur/). The authors gratefully
acknowledge OPUR partners (Ville de Paris, CD77, CD92, CD93, CD94, AESN, SIAAP) and Gustave Eiffel University for
their financial support.

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
