# Peer review of "Development and evaluation of a Sustainable Drainage System module into TEB (v 9.0) model"

_EGUsphere, 2025_

## Author Comment (AC1)

**Reviewer 1**

Dear Reviewer,

Thank you for your review, which will help us improve our article. Please find below a detailed response to each points you raised:

**Reviewing of the manuscript 'Development and evaluation of a Sustainable Drainage System module into TEB (V9.0) model' by Jose Manuel Tunqui Neira et al. (2025) submitted to Geoscientific Model Development (Manuscript ID: egusphere-2025-2831).**

**This work focuses on incorporate more elaborate urban drainage system SUDS module into the popular TEB model to better represent the hydrological processes under the combined climate change and anthropogenic effects. The authors have shown the equations and diagrams related to the SUDS module in a detailed and well represented way. The methodology and presentation are well defined, clearly stated, and properly validated. The work would be great beneficial to the hydro-climate modeling community. Therefore, I would recommend it be published after Minor Revision. The following are specific comments and suggestions that may help improve the manuscript quality:**

 **The reviewer would suggest reducing the amount of abbreviations in this section unless they are necessary. In addition, the abbreviation should be given its full name so that the readers are easy to follow, e.g., SWMM.**

We thank the reviewer for this suggestion. The Introduction has been revised to improve readability by reducing the number of acronyms. Non-essential abbreviations have been removed or replaced by their full names, and all remaining acronyms are now defined at their first occurrence (e.g., Storm Water Management Model (SWMM), Town Energy Balance (TEB), evapotranspiration (ET), and Interaction Soil–Biosphere–Atmosphere diffusion scheme (ISBA-DF)).

In addition, to further improve clarity and limit the repeated use of abbreviations and symbols in the main text, we have added a dedicated appendix (Appendix D) that summarizes all variables, fluxes and parameters involved in the hydrological formulation of the $E - SUDSa$ module (Eqs. 1–27), together with their definitions and physical units.

This appendix provides a centralized reference for notation, allowing the main text to remain more readable and less overloaded with symbols, while ensuring transparency and consistency in the model formulation.

1. **Starting from this Section and the following Sections, the authors please consider keeping some space at the start of the new paragraph, or keeping**

**some line space between the last paragraph and the following one. By doing so, the readers could better follow the meaning of the represented contents in a coherent way.**

The manuscript was initially formatted using the journal's standard template. However, following the reviewer's recommendation, additional spacing between paragraphs has been introduced where appropriate to improve readability and visual clarity.

2. **Line 21: '…… SUDS are ……' may change to '…… SUDS is ……'.**

This has been corrected in the revised manuscript.

3. **Line 44: '…… SUDS model' may change to 'SUDS module ……'.**

This has been corrected in the revised manuscript.

4. **Line 61: Please add the full name of TEB, I guess 'Town Energy Balance', when it is first appeared.**

The full name "Town Energy Balance (TEB)" has been added at its first occurrence.

5. **Line 71: 'The project's objective …..'. In this manuscript intending for publication, the authors may better focus on the manuscript or this article's objective.**

The project's objective" has been replaced by "the objective of this study ".

6. **Line 190: 'Table 1 …… Figure 1 [and] Figure 2'.**

The cross-references to Table 1, Figure 1 and Figure 2 have been corrected and clarified in the revised manuscript.

7. **Line 197: '…… portion of the mesh (Figure A.1 in Appendix A) ……'. Adding a bracket before 'Figure A.1'.**

The missing opening bracket has been added in the revised manuscript.

8. **Line 205: Is it proper to adding Figure 3 in the Section's title? Please consider revising it.**

We thank the reviewer for this comment and agree with the suggestion. The reference to Figure 3 has been removed from the section title. Instead, a reference to Figure 3 has been added in the first sentence of the section, indicating that the conceptual framework underlying all the processes described is presented in this figure. This revision improves consistency with standard manuscript structure and maintains clarity for the reader.

9. **The subsection's title may better be more detailed. For example, 'Section 4.4.1 Scenario 1 and Section 4.4.2 Scenario 2' could not provide sufficient information for the readers to follow this article and the following contents under these subsections. Please consider revise it.**

The subsection titles have been revised to be more informative. They now explicitly describe the purpose of each scenario, namely "Comparison of SWMM/LID and TEB / $E - SUDSa$ for a single facility" and "Combining multiple SUDS of the same type with different sizing configurations within the $E - SUDSa$."

10. **The percolations and exfiltration in Figure 5 and those corresponding yellow dots in Figure 7 show some abnormal results. For example, the E-SUDSSa at 25 mm/h while the LID-SWMM could range from 0-25 mm/h. Could you explain it and the similar issues in these two figures? Could the module be fixed to be better consistent between SWMM and TEB models for the SUDS hydrological processes?**

We thank the reviewer for highlighting this important point. In Figure 5, the differences observed between SWMM and TEB for percolation and exfiltration arise from the fundamentally different physical formulations used by the two models. SWMM applies a Green–Ampt infiltration scheme with a field-capacity threshold (Rossman and Huber, 2015), while TEB (via ISBA-DF) uses a free-drainage lower boundary condition that allows percolation whenever hydraulic conductivity permits (Albergel et al., 2018). This explanation has been added to the manuscript.

Figure 7 does not involve SWMM but compares aggregated TEB simulations (SUM-SUDS) with the equivalent $E - SUDS$ representation. The dispersion observed for percolation and exfiltration in this figure is mainly due to the averaging of substrate and storage parameters in the equivalent system, which modifies the timing and magnitude of vertical water transfers. This has also been clarified in the manuscript.

Because these differences originate from structural model assumptions and aggregation effects, forcing strict numerical equivalence would not be physically consistent; nevertheless, the overall dynamics and water balance remain in good agreement.

11. **The values at the left panels of Figure 6 and 8 may be changed to be in a more formal way.**

The numerical labels have been reformatted using standard notation instead of scientific "e" notation to improve readability.

12. **This part should be revised. Firstly, the future research direction may be switched to somewhere else. Then, the results of this work better be summarized in the form of listing bulletin points, e.g., (1) ....., (2), ..... (3)..... etc. Instead of put all together.**

The Conclusions section has been reorganized. The main results are now summarized as bullet points, and the future research perspectives have been moved to a separate final subsection.

13. **For the validation of the new module results, more statistical or skill metrics should be included, in addition to R2. For example, root-mean-square error, mean bias, relative bias, total sample number, and others etc. Please consider adding them to evaluate the model performance more carefully.**

We thank the reviewer for this important suggestion. The choice of statistical indicators in this study follows the conceptual framework of the two modelling scenarios.

In **Scenario 1**, SWMM and TEB represent two independent process-based models describing the same physical SUDS. In this context, SWMM is used as a reference model, as it is widely adopted by the urban hydrology community. However, the objective of this comparison is not to treat SWMM as observational truth, but to assess the consistency between two modelling approaches developed for different purposes. In particular, the TEB-SUDS module is designed to be applied at larger spatial scales, which implies some level of process aggregation, while relying on soil and surface parameterizations that are physically based and widely used in land-surface modelling (e.g. soil water transfers and evapotranspiration).

For this reason, classical performance metrics such as RMSE, mean bias or relative bias, which emphasize absolute deviations with respect to a reference, were complemented by symmetric indicators that focus on the similarity of temporal dynamics and variability. In addition to $R^2$, we therefore included the Pearson correlation coefficient (r), the standard deviation (σ)

Furthermore, because several hydrological processes are highly intermittent and characterized by a large proportion of zero values—primarily reflecting time steps with no rainfall and therefore no runoff generation—we introduced upper quantiles ($Q_{90}$ and $Q_{95}$) to specifically assess model behavior during active and high-flow periods, which are hydrologically the most relevant. Together, these indicators allow a balanced evaluation of agreement between SWMM and TEB without assuming that one model perfectly represents observed conditions. This information was added in sections 4.3 and 4.4.1 of the revised manuscript.

In **Scenario 2**, the objective is different: SUM-SUDS and E-SUDS are two representations within the same TEB modelling framework, where SUM-SUDS is

treated as an explicit reference solution. In this case, classical performance metrics such as the Nash–Sutcliffe efficiency (NSE), percent bias (PBIAS) and $R^2$ are appropriate to quantify how well the equivalent E-SUDS reproduces the aggregated SUM-SUDS behaviour. In addition, for consistency with Scenario 1, the Pearson correlation coefficient (r) has also been included to assess the similarity of temporal dynamics. We therefore consider that this set of indicators adequately addresses the reviewer's request for additional statistical metrics beyond $R^2$.

**Reviewer 2**

Dear Reviewer,

Thank you for your review, which will help us improve our article. Please find below a detailed response to each points you raised:

**The article presents a new module to simulate sustainable drainage system (SUDS) into the Town Energy Balance (TEB) model. The model simulations are compared to the simulations of other models or versions considered as reference under two scenarios.**

**The article is generally clear and well presented.**

**However one can regret that the evaluation remains in a pure modelling world (if I understood well), evaluating whether the proposed model is able to mimic the outputs of another model. Thus it is difficult to evaluate whether the proposed model can be trusted: the convergence with the outputs of the reference model only indicates that the two models compared are consistent, not that they are reliably simulating actual fluxes and states variables that could be observed in the real world. I am somewhat troubled by the fact that new models can be proposed in the literature without being tested against observations. And if there are no observations available to test these models, does that mean they cannot be falsified, in Popper's sense? This would be a problem I think. If complex models cannot be evaluated, how can we trust them? I find this is a strong limitation of this article. Or maybe I missed something.**

**This limitation should be more clearly stated, acknowledged and discussed in the abstract, discussion and conclusion of the paper. Else the reader may consider this as a proof of acceptability of the model, which I find it is not.**

We fully agree with the reviewer that observational evaluation is essential for establishing the real-world reliability of a model. Our long-term objective is indeed to confront the TEB-SUDS module with field measurements. However, the literature shows that long-term and process-resolved observational datasets for SUDS at the urban scale are extremely limited. Most studies focus on individual or small groups of devices, while catchment- or

city-scale effects are rarely observed directly (Golden and Hoghooghi, 2018). In addition, available monitoring data typically emphasizes runoff and peak flows, whereas other key processes such as groundwater recharge, baseflow and evapotranspiration are much less documented (Jefferson et al., 2017). Finally, monitoring and evaluation of SUDS and LID systems are known to suffer from important methodological and data limitations (Eckart et al., 2017).

Because of these constraints, we adopted in this study a model-to-model evaluation framework as a necessary first step to test the physical consistency and internal coherence of the new TEB-SUDS module under controlled conditions, before observational validation becomes feasible. For this reason, we deliberately use the term evaluation rather than validation throughout the manuscript.

We have now explicitly clarified this limitation and the scope of the present work in the Abstract, Discussion and Conclusions, to avoid any confusion between internal consistency and real-world evaluation.

**Some results shown in the article also exhibit unexpected behaviour (at least for me) that could be further analysed.**

**Detailed comments**

1. **Abstract: As mentioned in my comments above, the abstract should more clearly state that the evaluation is limited to a synthetic world. Lines 11 or 14 oversell model results and are misleading in that sense.**

   We thank the reviewer for this important remark. The Abstract has been revised to avoid any overstatement of model performance and to clarify the scope of the evaluation. It now explicitly indicates that the assessment is based on inter-model comparisons conducted within a synthetic modelling framework. In addition, the Abstract highlights that differences between SWMM and TEB are observed for subsurface processes such as percolation and exfiltration, which are linked to distinct representations of soil–water exchanges, while the overall hydrological behaviour and water balance remain physically consistent. This revised wording better reflects the nature and scope of the results presented in the manuscript.

2. **Section 3: It would be useful to have a table summarizing all the state variables and parameters used in Eqs. 1 to 27 (maybe in the appendix), with units, ranges, etc. This would be helpful for the reader (at least for me).**

   We thank the reviewer for this very helpful suggestion.

   To improve readability and facilitate interpretation of the hydrological formulation presented in Section 3, we have added a comprehensive summary table listing all variables, fluxes and parameters involved in Eqs. (1–27).

The new table (Appendix D, Table D1) explicitly distinguishes between:

(i)     state variables and fluxes computed internally by the model,
(ii)    model parameters prescribed as inputs, and
(iii)   hydroclimatic forcing variables provided as time series.

For each quantity, the table provides a concise definition and the associated physical unit. For parameters, indicative physical bounds are given when meaningful, while for internally computed variables no range is prescribed, as their values emerge from the model equations and boundary conditions.

Given its length and technical nature, the table has been placed in a dedicated appendix, and a reference to Appendix D has been added at the end of Section 3. We believe that this addition substantially improves the transparency, clarity and reproducibility of the proposed $E-SUDSa$ formulation.

3. **Section 4: I did not find how model warm-up was done. Incorrect model initialization may create strong modelling errors. This should be clarified.**

We thank the reviewer for pointing out this important issue. In the original version of the manuscript, the model warm-up procedure was not explicitly described. In line with standard practice for the TEB model, a spin-up period of four months was applied to allow soil moisture, storage and energy state variables to reach dynamic equilibrium before the evaluation period. We have now clarified this in Section 4.

For consistency, the same warm-up strategy has also been applied to the SWMM simulations, so that both models are initialized under comparable conditions prior to the analysis period. This information has been added to the revised manuscript. (Section 4.1.)

4. **Eq. 33: PBIAS criterion is often defined as the opposite (sim – ref). Please clarify what was actually used here and check criteria were consistently computed**

We thank the reviewer for pointing out this mistake. We acknowledge that the sign convention of PBIAS in Eq. (33) was incorrectly written in the original manuscript. We have carefully checked all PBIAS values reported in the manuscript and confirmed that they were computed using the correct formulation. Only the equation was incorrect and has now been revised accordingly.

5. **Fig. 5: There are strange behaviours on some graphs with accumulations of points around some specific values (mostly horizontal or vertical structures). It means that one simulation is almost constant when the other is not. Where does this come from? Is it expected and realistic? Were corresponding time series checked visually to better understand these behaviours?**

We thank the reviewer for this careful observation. These horizontal and vertical structures in Figure 5 are indeed caused by threshold-controlled processes and different conceptualizations of soil–water exchanges in SWMM and TEB. In particular, SWMM initiates percolation from the substrate into the storage layer only when soil moisture reaches field capacity (Rossman and Huber, 2015), whereas TEB applies a free-drainage lower boundary condition (Albergel et al., 2018), allowing percolation whenever the hydraulic conductivity of the bottom soil layer permits (Section 4.5.1; Fig. 5). This produces plateaus in one model while the other continues to evolve, resulting in vertical or horizontal clusters in the scatter plots.

Similar effects occur for surface storage and overflow, because in SWMM overflow is computed after infiltration and evapotranspiration, whereas in TEB overflow is computed before these processes (Eq. 1), which can lead to quasi-constant surface storage in one model while the other varies.

The corresponding time series were checked visually, and these structures were found to coincide with periods when soil moisture, surface storage or drainage processes reached physical thresholds (e.g. field capacity, reservoir capacity, or soil drainage limits). These behaviours are therefore expected and physically consistent with the respective model formulations, rather than numerical artefacts.

6. **Fig. 7: Same comment. Here there are also graphical structures which are not vertical, but showing some apparent thresholds. How can this be explained? In such cases of discrepancy, probably at least one model is strongly false though it is not possible to say which one. So how can we conclude in these cases. This is a real problem for the evaluation process presented here.**

We thank the reviewer for this important comment. Figure 7 does not compare two independent models, but two representations of the same TEB system: the explicit SUM-SUDS configuration and the equivalent $E-SUDS$ representation. The apparent threshold-like structures arise from the aggregation procedure itself. In the $E-SUDS$ approach, heterogeneous substrate and storage properties of multiple SUDS are averaged into a single equivalent system, which modifies the timing and magnitude of vertical water transfers compared with the explicit SUM-SUDS configuration.

As a result, small differences in soil moisture or storage state can be amplified when mapped into the equivalent representation, leading to non-linear responses and apparent thresholds in the scatter plots. These patterns therefore reflect the loss of fine-scale heterogeneity inherent to any aggregation procedure.

In this context, the objective of Figure 7 is not to identify a "true" and a "false" model, but to assess whether the equivalent $E-SUDS$ preserves the dominant hydrological behaviour and water balance of the explicit SUM-SUDS system. This is why the evaluation in Scenario 2 is based on consistency of fluxes and cumulative water balance rather than on point-wise statistical agreement.

7. **Building on the two previous comments, are there specific conditions where the two simulations differ most?**

   Yes. The largest differences between the simulations occur under conditions where threshold-controlled processes dominate the system response. In particular, discrepancies are most pronounced during periods of near-saturation or low-infiltration capacity, when small differences in soil moisture, substrate properties or drainage formulations can lead to strongly different percolation, exfiltration and overflow responses.

   In Scenario 1, this mainly corresponds to periods when soil moisture approaches field capacity or when exfiltration is activated, because SWMM and TEB apply different lower-boundary and infiltration formulations. In Scenario 2, the largest discrepancies occur when heterogeneous SUDS units with contrasting hydraulic properties are aggregated into a single equivalent system, which modifies the timing and magnitude of vertical fluxes. This explanation has been added to the manuscript.

8. **Discussion and conclusion: See major comment above.**

   In line with the major comment, both the Discussion and the Conclusion have been revised to more clearly state the scope and limitations of the present work. We now explicitly clarify that the evaluation relies on inter-model comparisons conducted in a synthetic modelling framework and does not yet constitute an observational evaluation.

   In addition, the Discussion and Conclusion now emphasize that the differences observed between SWMM and TEB for percolation and exfiltration processes do not indicate a deficiency of the $E-SUDSa$ module itself but rather arise from distinct representations of soil–water exchanges and drainage processes in the two models. These differences are inherent to the respective model formulations.

   Finally, the expected limitations of the equivalent $E-SUDS$ approach are now explicitly discussed. We highlight that the aggregation of heterogeneous SUDS into a single equivalent system may induce non-linear responses and additional uncertainty, and that this approach is primarily intended for applications at larger spatial scales. The need to further investigate conditions of applicability and to characterize the uncertainties induced by aggregation is identified as an important direction for future work.

**Authors' references**

Albergel, C., Boone, A., Belamari, S., Decharme, B., Dumont, M., Le Moigne, P., and Masson, V.: SURFEX V8.1. Scientific Documentation, 2018.

Eckart, K., McPhee, Z., and Bolisetti, T.: Performance and implementation of low impact development – A review, Sci. Total Environ., 607–608, 413–432, https://doi.org/10.1016/j.scitotenv.2017.06.254, 2017.

Golden, H. E. and Hoghooghi, N.: Green infrastructure and its catchment-scale effects: an emerging science, WIREs Water, 5, e1254, https://doi.org/10.1002/wat2.1254, 2018.

Jefferson, A. J., Bhaskar, A. S., Hopkins, K. G., Fanelli, R., Avellaneda, P. M., and McMillan, S. K.: Stormwater management network effectiveness and implications for urban watershed function: A critical review, Hydrol. Process., 31, 4056–4080, https://doi.org/10.1002/hyp.11347, 2017.

Rossman, L. A. and Huber, W.: Storm Water Management Model Reference Manual Volume III – Water Quality, US EPA Office of Research and Development, Washington, DC, 2015.